# Addressing Loss of Plasticity and Catastrophic Forgetting in Continual Learning

**Mohamed Elsayed**
Department of Computing Science
University of Alberta
Alberta Machine Intelligence Institute
mohamedelsayed@ualberta.ca

**A. Rupam Mahmood**
Department of Computing Science
University of Alberta
CIFAR Canada AI Chair, Amii
armahmood@ualberta.ca

## Abstract

Deep representation learning methods struggle with continual learning, suffering from both catastrophic forgetting of useful units and loss of plasticity, often due to rigid and unuseful units. While many methods address these two issues separately, only a few currently deal with both simultaneously. In this paper, we introduce Utility-based Perturbed Gradient Descent (UPGD) as a novel approach for the continual learning of representations. UPGD combines gradient updates with perturbations, where it applies smaller modifications to more useful units, protecting them from forgetting, and larger modifications to less useful units, rejuvenating their plasticity. We use a challenging streaming learning setup where continual learning problems have hundreds of non-stationarities and unknown task boundaries. We show that many existing methods suffer from at least one of the issues, predominantly manifested by their decreasing accuracy over tasks. On the other hand, UPGD continues to improve performance and surpasses or is competitive with all methods in all problems. Finally, in extended reinforcement learning experiments with PPO, we show that while Adam exhibits a performance drop after initial learning, UPGD avoids it by addressing both continual learning issues.[1]

## 1 Challenges of Continual Learning

Continual learning remains a significant hurdle for artificial intelligence, despite advancements in natural language processing, games, and computer vision. *Catastrophic forgetting* (McCloskey & Cohen 1989, Hetherington & Seidenberg 1989) in neural networks is widely recognized as a major challenge of continual learning (De Lange et al. 2021). The phenomenon manifests as the failure of gradient-based methods like SGD or Adam to retain or leverage past knowledge due to forgetting or overwriting previously learned units (Kirkpatrick et al. 2017). In continual learning, these learners often relearn recurring tasks, offering little gain over learning from scratch (Kemker et al. 2018). This issue also raises a concern for reusing large practical models, where finetuning them for new tasks causes significant forgetting of pretrained models (Chen et al. 2020, He et al. 2021).

Methods for mitigating catastrophic forgetting are primarily designed for specific settings. These include settings with independently and identically distributed (i.i.d.) samples, tasks fully contained within a batch or dataset, growing memory requirements, known task boundaries, storing past samples, and offline evaluation. Such setups are often impractical in situations where continual learning is paramount, such as on-device learning. For example, retaining samples may not be possible due to the limitation of computational resources (Hayes et al. 2019, Hayes et al. 2020, Hayes & Kannan 2022, Wang et al. 2023) or concerns over data privacy (Van de Ven et al. 2020).

In the challenging and practical setting of *streaming learning*, catastrophic forgetting is more severe and remains largely unaddressed (Hayes et al. 2019). In streaming learning, samples are presented to the learner as they arise, which is non-i.i.d. in most practical problems. The learner cannot retain the sample and is thus expected to learn from it immediately. Moreover, evaluation happens online on the most recently presented sample. This setup mirrors animal learning (Hayes et al. 2021, also c.f., list-learning, Ratcliff 1990) and is practical for many applications, such as robotics or autonomous on-device learning. In this work, we consider streaming learning with unknown task boundaries.

---

[1]Code is available at https://github.com/mohmdelsayed/upgd

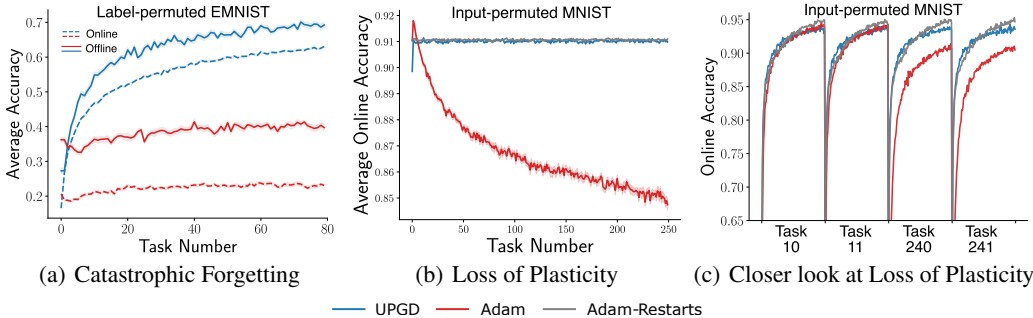

Figure 1: (a) Adam suffers from catastrophic forgetting and hence hardly improves performance. (b & c) Adam loses plasticity as newer and newer tasks are presented and performs much worse than Adam with restarts later. In contrast, our proposed method, UPGD, quickly learns and maintains plasticity throughout learning. See Appendix I.1 for experimental details.

Streaming learning provides the learner with a stream of samples $(\boldsymbol{x}_t, \boldsymbol{y}_t)$ generated using a non-stationary *target function* $f_t$ such that $\boldsymbol{y}_t = f_t(\boldsymbol{x}_t)$. The learner observes the input $\boldsymbol{x}_t \in \mathbb{R}^d$, outputs the prediction $\hat{\boldsymbol{y}}_t \in \mathbb{R}^m$, and then observes the true output $\boldsymbol{y}_t \in \mathbb{R}^m$, strictly in this order. The learner is then evaluated immediately based on the online metric $E(\boldsymbol{y}_t, \hat{\boldsymbol{y}}_t)$, for example, accuracy in classification or squared error in regression. The learner uses a neural network for prediction and $E$ or a related loss to learn the network parameters immediately without storing the sample. The target function $f_t$ is locally stationary, where changes to $f_t$ occur occasionally, creating a nonstationary continual learning problem composed of a sequence of stationary tasks.

Fig. 1(a) illustrates catastrophic forgetting with Adam in the streaming learning setting. Here, a sequence of tasks based on *Label-Permuted EMNIST* is presented to the learner. The tasks are designed to be highly coherent, where the features learned in one task are fully reusable in the other. Full details of the problem are described in Section 4.4. If the learner can remember and leverage prior learning, it should continue to improve performance as more tasks are presented. However, Fig. 1(a) reveals that Adam can hardly improve its performance, which remains at a low level of accuracy, indicating forgetting and relearning. Although catastrophic forgetting is commonly studied under offline evaluation (solid lines), the issue also manifests in online evaluation (dashed lines). This result indicates that current representation learning methods are unable to leverage previously learned useful features but instead forget and relearn them in subsequent tasks.

Yet another pernicious challenge of continual learning is *loss of plasticity*, where the learner's ability to learn new things diminishes. Recent studies reveal that SGD or Adam continues to lose plasticity with more tasks, primarily due to features becoming difficult to modify (Dohare et al. 2021, Lyle et al. 2023). Several methods exist to maintain plasticity, but they generally do not address catastrophic forgetting. Figures 1(b) and 1(c) illustrate loss of plasticity, where Adam is presented with a sequence of new tasks based on *Input-Permuted MNIST* (Goodfellow et al. 2013). Adam's performance degrades with more tasks and becomes worse than *Adam-Restarts*, which learns from scratch on each task. The stable performance of Adam-Restarts indicates that the tasks are of similar difficulty. Yet, Adam becomes slower to learn over time, demonstrating loss of plasticity.

A method that preserves useful units, such as features or weights, while leaving the other units adaptable would potentially address both catastrophic forgetting and loss of plasticity. Although a few methods address both issues simultaneously, such methods expect known task boundaries, maintain a replay buffer, or require pretraining, which does not fit streaming learning (Hayes et al. 2022). In this paper, we intend to fill this gap and present a continual learning method that addresses both catastrophic forgetting and loss of plasticity in streaming learning without such limitations.

## 2 RELATED WORKS

**Addressing Catastrophic Forgetting.** Different approaches have been proposed to mitigate catastrophic forgetting. For example, replay-based methods (e.g., Chaudhry et al. 2019, Isele & Cosgun 2018, Rolnick et al. 2019) address forgetting by using a replay buffer to store incoming non-i.i.d. data and then sample from the buffer i.i.d. samples. Catastrophic forgetting is also addressed by parameter isolation methods (e.g., Rusu et al. 2016, Schwarz et al. 2018, Wortsman et al. 2020, Ge et al. 2023) that can expand to accommodate new information without significantly affecting previously learned knowledge. There are also sparsity-inducing methods (e.g., Liu et al. 2019) that work

by maintaining sparse connections so that the weight updates can be localized and not affect many prior useful weights. Finally, regularization-based methods use a penalty to discourage the learner from moving too far from previously learned weights (Kirkpatrick et al. 2017, Aljundi et al. 2018, Aljundi et al. 2019) or approximations (Lan et al. 2023). The penalty amount is usually a function of the weight importance based on its contribution to previous tasks.

**Addressing Loss of Plasticity.** Dohare et al. (2023a) introduced a generate-and-test method (Appendix M, Mahmood & Sutton 2013) that maintains plasticity by continually replacing less useful features and showed that methods with continual injection of noise (e.g., Ash & Adams 2020) also maintain plasticity. Later, several methods were presented to retain plasticity. For example, Nikishin et al. (2023) proposed dynamically expanding the network, Abbas et al. (2023) recommended adapted activation functions, and Kumar et al. (2023) proposed regularization toward initial weights.

**Addressing Both Issues.** The trade-off between plasticity and forgetting has been outlined early by Carpenter & Grossberg (1987) as the stability-plasticity dilemma, a trade-off between maintaining performance on previous experiences and adapting to newer ones. The continual learning community focused more on improving the stability aspect by overcoming forgetting. Recently, however, there has been a new trend of methods that address both issues simultaneously. Chaudhry et al. (RWalk, 2018) utilized a regularization-based approach with a fast-moving average that quickly adapts to the changes in the weight importance, emphasizing the present and the past equally. Jung et al. (2022) introduced different techniques, including structure-wise distillation loss and pretraining to balance between plasticity and forgetting. Gurbuz et al. (2022) proposed using connection rewiring to induce plasticity in sparse neural networks. Finally, Kim et al. (2023) proposed using a separate network for learning the new task, which then is consolidated into a second network for previous tasks. Despite the recent advancement in addressing the two issues of continual learning, most existing methods do not fit the streaming learning setting since they require knowledge of task boundaries, replay buffers, or pretraining.

**Importance Measure in Neural Network Pruning.** Pruning in neural networks requires an importance metric to determine which weights to remove. Typically, the network is pruned using different measures such as the weight magnitude (e.g., Han et al. 2015, Park et al. 2020), first-order information (e.g., Mozer & Smolensky 1988, Hassibi & Stork 1992, Molchanov et al. 2016), second-order information (e.g., LeCun et al. 1989, Dong et al. 2017), or both (e.g., Tresp et al. 1996, Molchanov et al. 2019). Similar to pruning, regularization-based methods that address catastrophic forgetting use weight-importance measures such as the Fisher information diagonals to weigh their penalties.

## 3 METHOD

Our approach is to retain useful units while modifying the rest, which requires a metric to assess the utility or usefulness of the units. In this section, we introduce a measure for weight utility and outline an efficient method for computing it. Weight utility can be defined as the change in loss when setting the weight to zero, essentially removing its connection (Mozer & Smolensky 1988, Karnin 1990). Removing an important weight should result in increased loss. Ideally, both immediate and future losses matter, but we can only assess immediate loss at the current step. Finally, we devise a gradient-based update rule that protects or modifies weights based on this utility measure.

To define utility precisely, let us consider that the learner produces the predicted output $\hat{y}$ using a neural network with $L$ layers, parametrized by the set of weights $\mathcal{W} = \{\boldsymbol{W}_1, ..., \boldsymbol{W}_L\}$. Here $\boldsymbol{W}_l$ is the weight matrix at the $l$-th layer, and its element at the $i$-th row and the $j$-th column is denoted by $W_{l,i,j}$. At each layer $l$, we get the activation output $\boldsymbol{h}_l$ of the features by applying the activation function $\boldsymbol{\sigma}$ to the activation input $\boldsymbol{a}_l$: $\boldsymbol{h}_l = \boldsymbol{\sigma}(\boldsymbol{a}_l)$. We simplify notations by defining $\boldsymbol{h}_0 \doteq \boldsymbol{x}$. The activation output $\boldsymbol{h}_l$ is then multiplied by the weight matrix $\boldsymbol{W}_{l+1}$ of layer $l+1$ to produce the next activation input: $a_{l+1,i} = \sum_{j=1}^{d_l} W_{l+1,i,j} h_{l,j}, \forall i$, where $\boldsymbol{h}_l \in \mathbb{R}^{d_l}$. Here, $\boldsymbol{\sigma}$ applies activation element-wise for all layers except for the final layer, which becomes the softmax function.

The utility $U_{l,i,j}(Z)$ of the weight $i, j$ at layer $l$ and sample $Z$ is defined as

$$U_{l,i,j}(Z) \doteq \mathcal{L}(\mathcal{W}_{\neg[l,i,j]}, Z) - \mathcal{L}(\mathcal{W}, Z), \tag{1}$$

where $\mathcal{L}(\mathcal{W}, Z)$ is the sample loss given $\mathcal{W}$, and $\mathcal{L}(\mathcal{W}_{\neg[l,i,j]}, Z)$ is a counterfactual loss where $\mathcal{W}_{\neg[l,i,j]}$ is the same as $\mathcal{W}$ except the weight $W_{l,i,j}$ is set to 0. We refer to it as the *true utility*

to distinguish it from its approximations, which are referred to as either *approximated utilities* or simply utilities. Note that this utility is a global measure, and it provides a total ordering for weights according to their importance. However, computing it is prohibitive since it requires additional $N_w$ forward passes, where $N_w$ is the total number of weights.

### 3.1 SCALABLE APPROXIMATION OF THE TRUE UTILITY

Since the computation of the true utility is prohibitive, we aim to approximate it such that no additional forward passes are needed. To that end, we estimate the true utility by a second-order Taylor approximation. We expand the counterfactual loss $\mathcal{L}(\mathcal{W}_{\neg[l,i,j]}, Z)$ around the current weight $W_{l,i,j}$ and evaluate at weight zero. Hence, the quadratic approximation of $U_{l,i,j}(Z)$ can be written as

$$
\begin{aligned}
U_{l,i,j}(Z) &= \mathcal{L}(\mathcal{W}_{\neg[l,i,j]}, Z) - \mathcal{L}(\mathcal{W}, Z) \\
&\approx \mathcal{L}(\mathcal{W}, Z) + \frac{\partial \mathcal{L}(\mathcal{W}, Z)}{\partial W_{l,i,j}}(0 - W_{l,i,j}) + \frac{1}{2}\frac{\partial^2 \mathcal{L}}{\partial W_{l,ij}^2}(0 - W_{l,i,j})^2 - \mathcal{L}(\mathcal{W}, Z) \\
&= -\frac{\partial \mathcal{L}(\mathcal{W}, Z)}{\partial W_{l,i,j}}W_{l,i,j} + \frac{1}{2}\frac{\partial^2 \mathcal{L}(\mathcal{W}, Z)}{\partial W_{l,i,j}^2}W_{l,i,j}^2.
\end{aligned}
\tag{2}
$$

We refer to the utility measure containing the first term as the *first-order utility* and the measure containing both terms as the *second-order utility*. The computation required for the second-order term has quadratic complexity. Therefore, we use the approximation by Elsayed and Mahmood (2022) that provides a Hessian diagonal approximation in linear complexity. This makes the computation of both of our utility approximations linear in complexity and therefore scalable. Moreover, we present a way for propagating our approximated utilities by the *utility propagation theorem* in Appendix B. We also define the utility of a feature and provide its scalable approximation in Appendix C and D.

### 3.2 UTILITY-BASED PERTURBED GRADIENT DESCENT (UPGD)

Now, we devise a new method called *Utility-based Perturbed Gradient Descent* (UPGD) that performs gradient-based learning guided by utility-based information. The utility information is used as a gate, referred to as *utility gating*, for the gradients to prevent large updates to already useful weights, addressing forgetting. On the other hand, the utility information helps maintain plasticity by perturbing unuseful weights which become difficult to change through gradients (see Dohare et al. 2023a). The update rule of UPGD is given by

$$
w_{l,i,j} \leftarrow w_{l,i,j} - \alpha\left(\frac{\partial \mathcal{L}}{\partial w_{l,i,j}} + \xi\right)\left(1 - \bar{U}_{l,i,j}\right),
\tag{3}
$$

where $\xi \sim \mathcal{N}(0, 1)$ is noise, $\alpha$ is the step-size parameter, and $\bar{U}_{l,i,j} \in [0, 1]$ is a scaled utility. For important weights with utility $\bar{U}_{l,i,j} = 1$, the weight remains unaltered even by gradient descent, whereas unimportant weights with $\bar{U}_{l,i,j} = 0$ get updated by both perturbation and gradient descent.

Another variation of UPGD, which we call *non-protecting UPGD*, is to add the utility-based perturbation to the gradient as $w_{l,i,j} \leftarrow w_{l,i,j} - \alpha[\partial \mathcal{L}/\partial w_{l,i,j} + \xi(1 - \bar{U}_{l,i,j})]$. However, such an update rule can only help against loss of plasticity, not catastrophic forgetting, as useful weights are not protected from change by gradients. We include non-protecting UPGD in our experiments to validate that using the utility information as a gate for both the perturbation and the gradient update is necessary to mitigate catastrophic forgetting. We provide convergence analysis for both UPGD and Non-protecting UPGD on non-convex stationary problems in Appendix A.

Utility scaling is important for the UPGD update rule. We present here a global scaling and present a local scaling variation in Appendix E. The global scaled utility requires the maximum utility of all weights (e.g., instantaneous or trace) at every time step, which is given by $\bar{U}_{l,i,j} = \phi(U_{l,i,j}/\eta)$. Here $\eta$ is the maximum utility of the weights and $\phi$ is the scaling function, for which we use sigmoid. We show the pseudo-code of our method using the global scaled utility in Algorithm 1, where $\boldsymbol{F}_l$ contains first derivatives and $\boldsymbol{S}_l$ contains second-derivative approximations (see Appendix F for more details on the `GetDerivatives` function). We focus here on weight-wise UPGD and provide similar pseudo-codes for feature-wise UPGD in Appendix E.

UPGD update rule can be related to some existing update rules. When we perturb all weights evenly, that is, when all scaled utilities are zero, UPGD reduces to a well-known class of algorithms

called *Perturbed Gradient Descent* (PGD) (Zhou et al. 2019). The PGD learning rule is given by $w_{l,i,j} \leftarrow w_{l,i,j} - \alpha \left[ \partial \mathcal{L} / \partial w_{l,i,j} + \xi \right]$. It has been shown that a PGD with weight decay algorithm, known as *Shrink and Perturb* (S&P) (Ash & Adams 2020), can help maintain plasticity in continual classification problems (Dohare et al. 2023a) since maintaining small weights prevents weights from over-committing, making them easy to change. The learning rule of Shrink and Perturb can be written as $w_{l,i,j} \leftarrow \rho w_{l,i,j} - \alpha \left( \partial \mathcal{L} / \partial w_{l,i,j} + \xi \right)$, where $\rho = 1 - \lambda \alpha$ and $\lambda$ is the weight decay factor. When no noise is added, the update reduces to SGD with weight decay (Loshchilov & Hutter 2019), known as SGDW. Incorporating the useful role of weight decay into UPGD, we can write the *UPGD with weight decay* (UPGD-W) update rule as $w_{l,i,j} \leftarrow \rho w_{l,i,j} - \alpha \left( \frac{\partial \mathcal{L}}{\partial w_{l,i,j}} + \xi \right) \left( 1 - \bar{U}_{l,i,j} \right)$.

## 3.3 Forgetting and Plasticity Evaluation Metrics

Here, we present two metrics to characterize plasticity and forgetting in streaming learning. First, we introduce a new online metric to quantify plasticity. Neuroplasticity is usually defined as the ability of biological neural networks to change in response to some stimulus (Konorski 1948, Hebb 1949). Similarly, in artificial neural networks, plasticity can be viewed as the ability of a neural network to change its predictions in response to new information (Lyle et al. 2023). We provide a definition that captures the existing intuition in the literature. We define the plasticity of a learner given a sample as the ability to change its prediction to match the target. The learner achieves plasticity of 1 given a sample if it can exactly match the target and achieves plasticity of 0 if it achieves zero or negative progress toward the target compared to its previous prediction given the same sample. Formally, we define the sample plasticity to be $p(Z) = \max \left( 1 - \frac{\mathcal{L}(\mathcal{W}^\dagger, Z)}{\max(\mathcal{L}(\mathcal{W}, Z), \epsilon)}, 0 \right) \in [0, 1]$, where $\mathcal{W}^\dagger$ is the set of weights after performing the update and $\epsilon$ is a small number to maintain numeri-

---

**Algorithm 1** UPGD

Given a stream of data $\mathcal{D}$, a network $f$ with weights $\{ \boldsymbol{W}_1, ..., \boldsymbol{W}_L \}$.
Initialize step size $\alpha$, utility decay rate $\beta$, and noise standard deviation $\sigma$.
Initialize $\{ \boldsymbol{W}_1, ..., \boldsymbol{W}_L \}$.
Initialize $\boldsymbol{U}_l, \forall l$ and time step $t$ to zero.
**for** $(\boldsymbol{x}, \boldsymbol{y})$ in $\mathcal{D}$ **do**
  $t \leftarrow t + 1$
  **for** $l$ in $\{ L, L-1, ..., 1 \}$ **do**
    $\eta \leftarrow -\infty$
    $\boldsymbol{F}_l, \boldsymbol{S}_l \leftarrow$ GetDerivatives$(f, \boldsymbol{x}, \boldsymbol{y}, l)$
    $\boldsymbol{M}_l \leftarrow {}^1\!/{}_2 \boldsymbol{S}_l \circ \boldsymbol{W}_l^2 - \boldsymbol{F}_l \circ \boldsymbol{W}_l$
    $\boldsymbol{U}_l \leftarrow \beta \boldsymbol{U}_l + (1 - \beta) \boldsymbol{M}_l$
    $\hat{\boldsymbol{U}}_l \leftarrow \boldsymbol{U}_l / (1 - \beta^t)$
    **if** $\eta < \max(\hat{\boldsymbol{U}}_l)$ **then** $\eta \leftarrow \max(\hat{\boldsymbol{U}}_l)$
  **for** $l$ in $\{ L, L-1, ..., 1 \}$ **do**
    Sample $\boldsymbol{\xi}$ elements from $\mathcal{N}(0, \sigma^2)$
    $\bar{\boldsymbol{U}}_l \leftarrow \phi(\hat{\boldsymbol{U}}_l / \eta)$
    $\boldsymbol{W}_l \leftarrow \boldsymbol{W}_l - \alpha (\boldsymbol{F}_l + \boldsymbol{\xi}) \circ (1 - \bar{\boldsymbol{U}}_l)$

---

cal stability. Note that the term $\left( 1 - \frac{\mathcal{L}(\mathcal{W}^\dagger, Z)}{\max(\mathcal{L}(\mathcal{W}, Z), \epsilon)} \right) \in (-\infty, 1]$ has an upper bound of 1, since $\mathcal{L}(\mathcal{W}, Z) \in [0, \infty), \forall \mathcal{W}, Z$ for cross-entropy and squared-error losses. We use this metric to measure plasticity directly, especially since most measures that are introduced (e.g., weight norm) to show loss of plasticity do not often correlate with plasticity (see Lyle et al. 2023). Our metric can be viewed as a baseline-normalized version of the plasticity metric by Lyle et al. (2023), where the baseline is the loss prior to the update. We measure loss of plasticity as $\Delta \bar{p}_{k+1} = \bar{p}_k - \bar{p}_{k+1}$, where $\bar{p}_k$ is the average plasticity over all samples in the $k$-th evaluation window. Note that this metric ranges from $-1$ to 1, $\Delta \bar{p}_k \in [-1, 1], \forall k$. Negative values indicate the learner has gained plasticity, whereas positive values indicate the learner has lost plasticity. In our experiments, we report the overall loss of plasticity on $T$ evaluation windows: $\sum_{k=1}^{T-1} \Delta \bar{p}_{k+1} = \bar{p}_1 - \bar{p}_T$.

Existing metrics for catastrophic forgetting are predominantly based on offline evaluations. In streaming learning, forgetting previously learned useful features leads to future learning deterioration rather than affecting past performance. If the learner keeps improving its representations with each task, the average online accuracy on new tasks should improve continually. On the other hand, if it cannot improve representations, its average online accuracy on new tasks may stay the same or even decrease (see Fig. 1(a)). Hence, we propose measuring forgetting using different online evaluation windows for streaming learning. Inspired by the windowed-forgetting metric introduced by De Lange et al. (2023), we propose the metric $F_{k+1} = A_k - A_{k+1}$, where $A_k$ is the accuracy averaged over all samples on the $k$-th evaluation window. This metric assumes the learned representations in one task remain relevant for future tasks, and all tasks have the same complexity. Note that this metric ranges from $-1$ to 1, $F_k \in [-1, 1], \forall k$, where negative values indicate the learner can improve on previous representations, and positive values indicate re-learning, hence forgetting. In our experiments, we report the overall forgetting on $T$ evaluation windows, $\sum_{k=1}^{T-1} F_{k+1} = A_1 - A_T$.

## 4    EXPERIMENTS

In this section, we begin by studying the quality of our approximated utilities. Then we study the effectiveness of UPGD in mitigating loss of plasticity and catastrophic forgetting. For the latter, we use non-stationary streaming problems based on MNIST (LeCun et al. 1998), EMNIST (Cohen et al. 2017), CIFAR-10 (Krizhevsky 2009), and ImageNet (Deng et al. 2009) datasets with learners that use multi-layer perceptrons, convolutional neural networks (LeCun et al. 1998), and residual neural networks (He et al. 2016) (however, see Appendix G for validating UPGD on stationary tasks). We also validate UPGD in extended reinforcement learning experiments. UPGD is compared to suitable baselines for our streaming learning setup, that is, without replay, batches, or task boundaries.

The performance of continual learners is evaluated based on the average online accuracy for classification problems. In each of the following experiments, a thorough hyperparameter search is conducted (see Appendix I). Our criterion was to find the best set of hyperparameters for each method that maximizes the area under the online accuracy curve. Unless stated otherwise, we averaged the performance of each method over 20 independent runs. We focus on the key results here and give the full experimental details in Appendix I.

### 4.1    QUALITY OF THE APPROXIMATED UTILITIES

A high-quality approximation of utility should give a similar ordering of weights to the true utility. We use the ordinal correlation measure of Spearman to quantify the quality of our utility approximations. An SGD learner with a small neural network with ReLU activations is used on a simple problem to minimize the online squared error.

At each time step, Spearman's correlation is calculated for first- and second-order global utility against the random ordering, the squared-gradient utility, and the weight-magnitude utility. We report the correlations between the true utility and approximated global weight utilities in Fig. 2. The correlation is the highest for the second-order utility throughout learning. On the other hand, the first-order utility becomes less correlated when the learner plateaus, likely due to zigzagging gradient el-

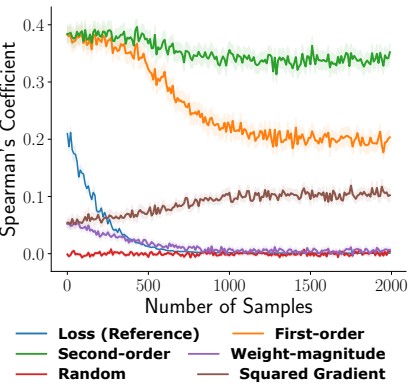

Figure 2: Rank correlation between the true utility and approximated utility.

ements near the solution. The weight-magnitude utility shows a small correlation to the true utility that gets smaller. The correlation of the squared-gradient utility increases with time steps but remains smaller than that of the first-order utility. We use random ordering as a baseline, which maintains zero correlation with the true utility, as expected. We also show the correlation between approximated local utility and true utility in Appendix J in addition to results with other activations.

### 4.2    UPGD AGAINST LOSS OF PLASTICITY

In this section, we use *Input-Permuted MNIST*, a problem where only loss of plasticity is present, and answer two main questions: 1) how UPGD and the other continual learning methods perform on this problem, and 2) whether performance alone is indicative of plasticity in this task. Performance has been used to measure plasticity on various problems and settings (Dohare et al. 2021, Nikishin et al. 2023, Kumar et al. 2023, Abbas et al. 2023). The decaying performance is usually attributed to loss of plasticity. Here, we question using performance to measure plasticity in arbitrary problems since performance can also be affected by other issues such as catastrophic forgetting (see Fedus et al. 2020). However, we hypothesize that if the only underlying issue is loss of plasticity, performance might actually reflect plasticity. Therefore, it is necessary to use a problem where only loss of plasticity is present to test our hypothesis. This approach also allows us to study the effectiveness of UPGD against loss of plasticity, isolated from catastrophic forgetting. In Input-permuted MNIST, we permute the inputs every 5000 steps where the time step at each permutation marks the beginning of a new task. After each permutation, the learned features become irrelevant to the new task, so the learner is expected to overwrite prior-learned representations as soon as possible. Thus, the input-permuted MNIST is a suitable problem to study loss of plasticity.

We compare SGDW, PGD, S&P, which addresses loss of plasticity, Adam with weight decay (Loshchilov & Hutter 2019) known as AdamW, UPGD-W, and Non-protecting UPGD-W. We also introduce and compare against *Streaming Elastic Weight Consolidation* (S-EWC), *Streaming Synaptic Intelligence* (S-SI), and *Streaming Memory-Aware Intelligence* (S-MAS). These methods can be viewed as a natural extension of EWC (Kirkpatrick et al. 2017), SI (Zenke et al. 2017), and MAS (Aljundi et al. 2018), respectively, which are regularization-based methods for mitigating forgetting to the streaming learning setting. Finally, we introduce and compare against *Streaming RWalk* (S-RWalk). This can be seen as a natural extension of RWalk (Chaudhry et al. 2018), a method that addresses both issues, adapted for streaming learning. We write the update rule of the last four methods in streaming learning as $w_{l,i,j} \leftarrow w_{l,i,j} - \alpha \left[ \partial \mathcal{L} / \partial w_{l,i,j} + \kappa \Omega_{l,i,j}(w_{l,i,j} - \bar{w}_{l,i,j}) \right]$, where $\kappa$ is the regularization factor, $\Omega_{l,i,j}$ is the weight importance, and $\bar{w}_{l,i,j}$ is the trace of weight $i, j$ at the $l$-th layer. The weight importance is estimated as a trace of squared gradients in S-EWC, whereas it is estimated as a trace of gradient magnitudes in S-MAS. S-RWalk is different from RWalk since it uses a trace of previous weights instead of the instantaneous previous weights. We omit the details of S-SI and RWalk weight importance estimation here and defer it to Appendix I.4. Lastly, since we compare against methods that use first-order information, we use global first-order utility traces in this and subsequent experiments for a fair comparison (however, see Appendix H for an experiment using second-order utilities).

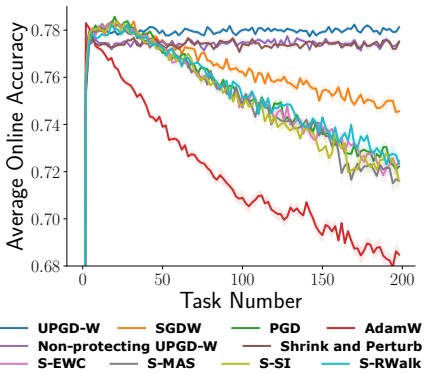

Figure 3: Performance of methods on the Input-permuted MNIST problem.

Fig. 3 shows that methods that only address catastrophic forgetting (e.g., S-EWC) continue to decay in performance whereas methods that address loss of plasticity alone (e.g., S&P) or together with catastrophic forgetting (e.g., UPGD), except S-RWalk, maintain their performance level. We plot the average online accuracy against the number of tasks. The average online accuracy is the percentage of correct predictions within each task, where the sample online accuracy is 1 for correct prediction and 0 otherwise. The prediction of the learner is given by argmax over its output probabilities. The learners are presented with a total of 1 million examples, one example per time step, and are required to maximize online accuracy using a multi-layer ($300 \times 150$) network with ReLU units.

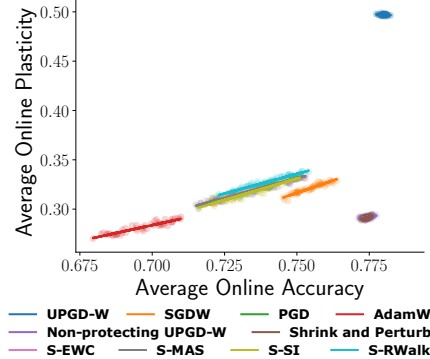

Figure 4: Each method's average plasticity against average accuracy of on Input-permuted MNIST.

In order to answer the second question, we measure online plasticity as defined in Section 3.3 for each method and plot performance against measured plasticity in Fig. 4. The average online plasticity of each method is shown against the average online accuracies of the last 100 tasks. We notice that plasticity and accuracy are strongly correlated, indicating that when the learner loses plasticity, its accuracy also decreases. This result corroborates that when the only underlying issue is loss of plasticity, performance indeed reflects plasticity. Note that such a correlation is not expected when loss of plasticity is not the only issue, and hence, the plasticity metric can be generally more reliable.

Finally, in Fig. 5, we use diagnostic statistics to further analyze the solutions each method achieves. More statistics for learners on this problem and next ones are reported in Appendix K.1. Notably, the results show that, for all methods, the fraction of zero activations increases and $\ell_0$ gradient norm decreases substantially except for UPGD-W along with Non-protecting UPGD-W and S&P.

## 4.3 UPGD Against Catastrophic Forgetting

Now, we study how UPGD and other continual learning methods address forgetting, and for that we use *Label-permuted CIFAR-10*. CIFAR-10 dataset contains 60,000 RGB images of size $32 \times 32$ belonging to 10 classes; each class has 6000 images. The labels are permuted every 2500 time step.

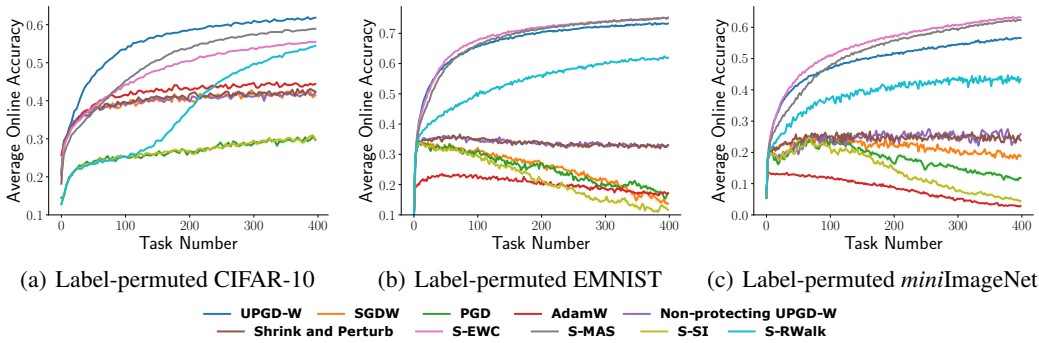

Figure 5: Diagnostic statistics on Input-permuted MNIST. The percentage of zero activations, $\ell_0$-norm and $\ell_1$-norm of the gradients, and $\ell_1$-norm of the weights are shown. We stacked the elements from the network gradients or weights into vectors to compute each norm at every sample.

Each learner is trained for 1M samples, one sample each time step, using a convolutional neural network with ReLU activations. Such permutations should not make the learner change its learned representations since it can simply change the weights of the last layer to adapt to that change. This makes the Label-permuted CIFAR10 problem suitable for studying catastrophic forgetting. It is not clear, however, whether the issue of loss of plasticity is present. We hypothesize that the issue of loss plasticity does not occur in this problem mainly due to its small number of classes (Lesort et al. 2023), leading to a probability of 10% for the same label re-occurrence after label permutation, resulting in less amount of non-stationarity that causes loss of plasticity.

| (a) Label-permuted CIFAR-10 | (b) Label-permuted EMNIST | (c) Label-permuted *mini*ImageNet |

Figure 6: Performance of methods on Label-permuted CIFAR-10, Label-permuted EMNIST, and Label-permuted mini-ImageNet. The higher the online accuracy, the better.

Fig. 6(a) shows that methods addressing catastrophic forgetting continually improve their performance. We use our loss of plasticity metric to check whether learners experience any loss of plasticity. Fig. 7(b) shows that the majority of methods have negative values, reflecting no loss of plasticity, which also indicates that catastrophic forgetting is the major issue in this problem. Although all learners can improve their performance, some can improve more than others, according to their forgetting metric (see Fig. 7(a)). We observe that learners without an explicit mechanism for addressing forgetting (e.g., AdamW) can reach a maximum accuracy of slightly above 40%, compared to learners addressing catastrophic forgetting that keep improving their performance.

## 4.4 UPGD AGAINST LOSS OF PLASTICITY AND CATASTROPHIC FORGETTING

In this section, we study the interplay of catastrophic forgetting and loss of plasticity using the *Label-permuted EMNIST* problem. The EMNIST dataset is an extended form of MNIST that has 47 classes, of both digits and letters, instead of just 10 digits. We permute the inputs every 2500 time steps and present the learners with a total 1 million examples, one example per time step, using the same network architecture from the first problem. Hence, this problem is also suitable for studying catastrophic forgetting. Since EMNIST has more classes, label re-occurrence probability becomes significantly smaller, leading to more non-stationarity. Thus, we expect that loss of plasticity might be present in this problem. Fig. 7(b) shows that most learners indeed suffer from loss of plasticity.

Fig. 6(b) shows that methods addressing catastrophic forgetting, including UPGD-W but except S-RWalk and S-SI, keep improving their performance and outperform methods that only address loss of plasticity. Notably, we observe that S-RWalk, which addressed catastrophic forgetting in the previous problem, struggles in this problem, likely due to the additional loss of plasticity. On the other hand, the performance of S-SI and the rest of the methods keeps deteriorating over time.

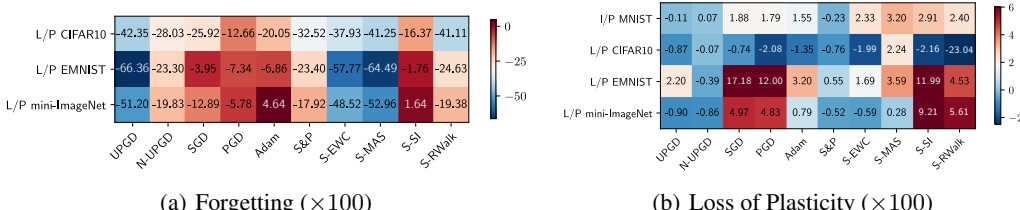

(a) Forgetting (×100)  (b) Loss of Plasticity (×100)

Figure 7: Forgetting and loss of plasticity of each method in different problems. I/P is short for input-permuted, and L/P is short for label-permuted. (a) forgetting is reported using our overall forgetting metric, where positive values indicate forgetting. The metric is computed starting from the sample number 1 using windows of twice the task length. (b) loss of plasticity is measured based on the overall loss in plasticity, meaning that positive values indicate loss of plasticity. The metric is computed starting from the sample number 0.5 million using windows of twice the task length.

Next, we perform a large-scale experiment using the *Label-permuted mini-ImageNet* problem, which has a large number of classes; hence, loss of plasticity is expected along with catastrophic forgetting. The mini-ImageNet (Vinyals et al. 2016) is a subset of the ImageNet dataset. The mini-ImageNet dataset contains $60,000$ RGB images of size $84 \times 84$ belonging to $100$ classes; each class has $600$ images. In Label-permuted mini-ImageNet, the labels are permuted every $2500$ time step. Each learner uses a fully connected network of two layers on top of a pre-trained *ResNet-50* (He et al. 2016) on ImageNet with fixed weights.

Fig. 6(c) exhibits the same trends manifested in the previous problem, where methods addressing catastrophic forgetting (e.g., S-EWC) performed the best, whereas methods addressing loss of plasticity (e.g., S&P) only maintained their performance at a lower level. Fig. 7(a) and Fig. 7(b) demonstrate that both metrics of UPGD-W are among the best overall values across all problems, indicating diminished forgetting and loss of plasticity. We refer the reader to Appendix L for an ablation study on the components of UPGD.

## 4.5 UPGD AGAINST POLICY COLLAPSE

Lastly, we study the role of UPGD in preventing a gradual performance drop or *policy collapse* (Dohare et al. 2023b) in reinforcement learning (RL). Dohare et al. (2023b) showed that non-stationarities in RL might exhibit both continual learning issues and demonstrated that the PPO algorithm (Schulman et al. 2017) can experience policy collapse when trained for an extended period. Since UPGD helps against catastrophic forgetting and loss of plasticity, we test whether UPGD can address policy collapse in PPO. We devise an adaptive variant of our method based on Adam normalization (Kingma & Ba 2015), which we call *Adaptive UPGD* (AdaUPGD), making its step size robust in several RL environments without tuning. We refer the reader to Algorithm 5 for the full description and Appendix I.6 for experimental details. Fig. 8 shows that AdaUPGD continually improves its performance, unlike Adam, which suffers from policy collapse.

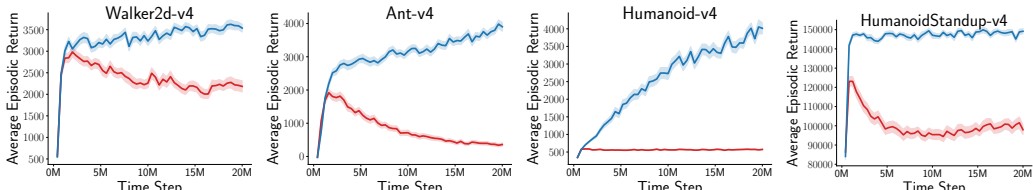

Figure 8: Performance of Adaptive UPGD (blue) and Adam (red) with PPO in different MuJoCo environments. The results are averaged over $40$ independent runs.

## 5 CONCLUSION

In this paper, we introduced a novel approach to mitigating loss of plasticity and catastrophic forgetting. We devised learning rules that protect useful weights and perturb less useful ones, thereby maintaining plasticity and reducing forgetting. We performed a series of challenging streaming learning experiments with many non-stationarities alongside reinforcement learning experiments. Our experiments showed that UPGD maintains network plasticity and reuses previously learned useful features, being among the only few methods that can address both issues effectively. Our work endeavors to pave the way for a new class of methods for addressing these issues together. Finally, we discuss the limitations of our method and future works in Appendix N.

ACKNOWLEDGEMENT

We gratefully acknowledge funding from the Canada CIFAR AI Chairs program, the Reinforcement Learning and Artificial Intelligence (RLAI) laboratory, the Alberta Machine Intelligence Institute (Amii), and the Natural Sciences and Engineering Research Council (NSERC) of Canada. We would also like to thank the Digital Research Alliance of Canada for providing the computational resources needed. We especially thank Shibhansh Dohare and Homayoon Farrahi for the useful discussions that helped improve this paper. We thank Nishanth Anand, Alex Lewandowski, and Khurram Javed for the helpful discussions. We also thank all of the Lifelong Reinforcement Learning Barbados Reinforcement Learning Workshop 2023 participants for their feedback and discussions.

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

# A  CONVERGENCE ANALYSIS FOR UPGD AND NON-PROTECTING UPGD

In this section, we provide convergence analysis for UPGD and Non-protecting UPGD in nonconvex stationary problems. We focus on the stochastic version of these two algorithms since we are interested in continual learners performing updates at every time step. The following proof shows the convergence to a stationary point up to the statistical limit of the variance of the gradients, where $\|\nabla f(\boldsymbol{\theta})\|^2 \leq \delta$ represent a $\delta$-accurate solution and is used to measure the stationarity of $\boldsymbol{\theta}$. Nonconvex optimization problems can be written as:

$$\min_{\boldsymbol{\theta} \in \mathbb{R}^d} f(\boldsymbol{\theta}) \doteq \mathbb{E}_{S \sim P}\left[\mathcal{L}(\boldsymbol{\theta}, S)\right],$$

where $f$ is the expected loss, $\mathcal{L}$ is the sample loss, $S$ is a random variable for samples and $\boldsymbol{\theta}$ is a vector of weights parametrizing $\mathcal{L}$. We assume that $\mathcal{L}$ is $L$-smooth, meaning that there exist a constant $L$ that satisfy

$$\|\nabla\mathcal{L}(\boldsymbol{\theta}_1, s) - \nabla\mathcal{L}(\boldsymbol{\theta}_2, s)\| \leq L\|\boldsymbol{\theta}_2 - \boldsymbol{\theta}_1\|, \ \forall \boldsymbol{\theta}_1, \boldsymbol{\theta}_2 \in \mathbb{R}^d, s \in \mathcal{S}. \tag{4}$$

We further assume that $\mathcal{L}$ has bounded variance in the gradients $\mathbb{E}[\|\nabla\mathcal{L}(\boldsymbol{\theta}, S) - \nabla f(\boldsymbol{\theta})\|^2] \leq \sigma^2, \forall \boldsymbol{\theta} \in \mathbb{R}^d$. Note that the assumption of L-smoothness on the sample loss result in L-smooth expected loss too, which is given by $\|\nabla f(\boldsymbol{\theta}_1) - \nabla f(\boldsymbol{\theta}_2)\| \leq L\|\boldsymbol{\theta}_1 - \boldsymbol{\theta}_2\|$. We assume that the perturbation noise in both UPGD versions has bounded variance $\mathbb{E}[\|\boldsymbol{\xi}\|^2] \leq \sigma_n^2$. For the simplicity of this proof, we use the true instantaneous weight utility, not an approximated one. We assume that the utility of any connection in the network is upper bounded by a number close to one $\bar{u} \to 1^-$.

## A.1  NON-PROTECTING UTILITY-BASED PERTURBED GRADIENT DESCENT

Remember that the update equation of the Non-protecting UPGD can be written as follows when the parameters are stacked in a single vector $\boldsymbol{\theta}$:

$$\theta_{t+1,i} = \theta_{t,i} - \alpha(g_{t,i} + \xi_{t,i}\rho_{t,i}).$$

where $\alpha$ is the step size, $\boldsymbol{g}_t$ is the sample gradient vector at time $t$, $\boldsymbol{\rho}_t = (1 - \boldsymbol{u}_t)$ is the opposite utility vector, and $\boldsymbol{\xi}_t$ is the noise perturbation. Since the function $f$ is $L$-smooth, we can write the following:

$$f(\boldsymbol{\theta}_{t+1}) \leq f(\boldsymbol{\theta}_t) + (\nabla f(\boldsymbol{\theta}_t))^\top(\boldsymbol{\theta}_{t+1} - \boldsymbol{\theta}_t) + \frac{L}{2}\|\boldsymbol{\theta}_{t+1} - \boldsymbol{\theta}_t\|_2^2 \tag{5}$$

$$= f(\boldsymbol{\theta}_t) - \alpha\sum_{i=1}^d \left(\nabla[f(\boldsymbol{\theta}_t)]_i(g_{t,i} + \xi_{t,i}\rho_{t,i})\right) + \frac{L\alpha^2}{2}\sum_{i=1}^d (g_{t,i} + \xi_{t,i}\rho_{t,i})^2. \tag{6}$$

Next, we take the conditional expectation of $f(\boldsymbol{\theta}_{t+1})$ as follows:

$$\mathbb{E}_t[f(\boldsymbol{\theta}_{t+1})|\boldsymbol{\theta}_t] \leq f(\boldsymbol{\theta}_t) - \alpha\sum_{i=1}^d \nabla[f(\boldsymbol{\theta}_t)]_i\mathbb{E}_t[(g_{t,i} + \xi_{t,i}\rho_{t,i})] + \frac{L\alpha^2}{2}\sum_{i=1}^d \mathbb{E}_t[(g_{t,i} + \xi_{t,i}\rho_{t,i})^2]$$

$$= f(\boldsymbol{\theta}_t) - \alpha\sum_{i=1}^d \nabla[f(\boldsymbol{\theta}_t)]_i\mathbb{E}_t[g_{t,i}] + \frac{L\alpha^2}{2}\sum_{i=1}^d \left(\mathbb{E}_t[g_{t,i}^2] + \mathbb{E}_t[(\xi_{t,i}\rho_{t,i})^2]\right)$$

$$= f(\boldsymbol{\theta}_t) - \alpha\sum_{i=1}^d \nabla[f(\boldsymbol{\theta}_t)]_i^2 + \frac{L\alpha^2}{2}\sum_{i=1}^d \left(\mathbb{E}_t[g_{t,i}^2] + \mathbb{E}_t[(\xi_{t,i}\rho_{t,i})^2]\right)$$

$$= f(\boldsymbol{\theta}_t) - \alpha\|\nabla f(\boldsymbol{\theta}_t)\|^2 + \frac{L\alpha^2}{2}\sum_{i=1}^d \left(\mathbb{E}_t[g_{t,i}^2] + \mathbb{E}_t[(\xi_{t,i}\rho_{t,i})^2]\right)$$

$$\leq f(\boldsymbol{\theta}_t) - \alpha\|\nabla f(\boldsymbol{\theta}_t)\|^2 + \frac{L\alpha^2}{2}\sum_{i=1}^d \mathbb{E}_t[g_{t,i}^2] + \frac{L\alpha^2}{2}\sigma_n^2.$$

Note that $\mathbb{E}_t[g_{t,i}] = [\nabla f(\boldsymbol{\theta}_t)]_i$, $\mathbb{E}_t[\xi_{t,i}] = 0$, and $\mathbb{E}[(\xi_{t,i}\rho_{t,i})^2] \leq \mathbb{E}[\xi_{t,i}^2]$, since $0 \leq \rho_{t,i} \leq 1 \; \forall t, i$. From the bounded variance assumption, we know that the $\mathbb{E}[\|\boldsymbol{g}_t\|^2]$ is bounded as follows:

$$\mathbb{E}[\|\boldsymbol{g}_t\|^2] \leq \frac{\sigma^2}{b_t} + \|\nabla f(\boldsymbol{\theta}_t)\|^2,$$

where $b_t$ is the batch size at time step $t$. We can now bound $\mathbb{E}_t[f(\boldsymbol{\theta}_{t+1})|\boldsymbol{\theta}_t]$ as follows:

$$\mathbb{E}_t[f(\boldsymbol{\theta}_{t+1})|\boldsymbol{\theta}_t] \leq f(\boldsymbol{\theta}_t) - \alpha\|\nabla f(\boldsymbol{\theta}_t)\|^2 + \frac{L\alpha^2}{2}\left(\sigma_n^2 + \frac{\sigma^2}{b_t} + \|\nabla f(\boldsymbol{\theta}_t)\|^2\right)$$

$$= f(\boldsymbol{\theta}_t) - \frac{2\alpha - L\alpha^2}{2}\|\nabla f(\boldsymbol{\theta}_t)\|^2 + \frac{L\alpha^2}{2}\left(\sigma_n^2 + \frac{\sigma^2}{b_t}\right).$$

Rearranging the inequality, taking expectations on both sides, and using the telescopic sum, we can write the following:

$$\frac{2\alpha - L\alpha^2}{2}\sum_{t=1}^{T}\mathbb{E}\|\nabla f(\boldsymbol{\theta}_t)\|^2 \leq f(\boldsymbol{\theta}_1) - \mathbb{E}[f(\boldsymbol{\theta}_{T+1})] + \frac{L\alpha^2 T}{2}\left(\sigma_n^2 + \frac{\sigma^2}{b_t}\right).$$

Multiplying both sides by $\frac{2}{T(2\alpha - L\alpha^2)}$ and using the fact that $f$ is the lowest at the global minimum $\boldsymbol{\theta}^*$: $f(\boldsymbol{\theta}_{T+1}) \geq f(\boldsymbol{\theta}^*)$, we can write the following:

$$\frac{1}{T}\sum_{t=1}^{T}\mathbb{E}\|\nabla f(\boldsymbol{\theta}_t)\|^2 \leq 2\frac{f(\boldsymbol{\theta}_1) - f(\boldsymbol{\theta}^*)}{T(2\alpha - L\alpha^2)} + \frac{L\alpha^2 2(\sigma_n^2 b_t + \sigma^2)}{b_t(2\alpha - L\alpha^2)}.$$

Therefore, the algorithm converges to a stationary point. However, in the limit $T \to \infty$, the algorithm has to have an increasing batch size or a decreasing step size to converge, which is the same requirement for convergence of other stochastic gradient-based methods at the limit (see Zaheer et al. 2018, Ammar 2020).

## A.2 UTILITY-BASED PERTURBED GRADIENT DESCENT

Remember that the update equation of UPGD can be written as follows when the parameters are stacked in a single vector $\boldsymbol{\theta}$:

$$\theta_{t+1,i} = \theta_{t,i} - \alpha(g_{t,i} + \xi_{t,i})\rho_{t,i}.$$

where $\alpha$ is the step size, $\boldsymbol{g}_t$ is the sample gradient vector at time $t$, $\boldsymbol{\rho}_t = (1 - \boldsymbol{u}_t)$ is the opposite utility vector, and $\boldsymbol{\xi}_t$ is the noise perturbation. Since the function $f$ is $L$-smooth, we can write the following:

$$f(\boldsymbol{\theta}_{t+1}) \leq f(\boldsymbol{\theta}_t) + (\nabla f(\boldsymbol{\theta}_t))^\top(\boldsymbol{\theta}_{t+1} - \boldsymbol{\theta}_t) + \frac{L}{2}\|\boldsymbol{\theta}_{t+1} - \boldsymbol{\theta}_t\|_2^2 \tag{7}$$

$$= f(\boldsymbol{\theta}_t) - \alpha\sum_{i=1}^{d}(\nabla[f(\boldsymbol{\theta}_t)]_i\rho_{t,i}(g_{t,i} + \xi_{t,i})) + \frac{L\alpha^2}{2}\sum_{i=1}^{d}(g_{t,i} + \xi_{t,i})^2\rho_{t,i}^2. \tag{8}$$

Next, we take the conditional expectation of $f(\boldsymbol{\theta}_{t+1})$ as follows:

$$\mathbb{E}_t[f(\boldsymbol{\theta}_{t+1})|\boldsymbol{\theta}_t] \leq f(\boldsymbol{\theta}_t) - \alpha\sum_{i=1}^{d}\nabla[f(\boldsymbol{\theta}_t)]_i\mathbb{E}_t[\rho_{t,i}(g_{t,i} + \xi_{t,i})] + \frac{L\alpha^2}{2}\sum_{i=1}^{d}\mathbb{E}_t[(g_{t,i} + \xi_{t,i})^2\rho_{t,i}^2]$$

$$= f(\boldsymbol{\theta}_t) - \alpha\sum_{i=1}^{d}\nabla[f(\boldsymbol{\theta}_t)]_i^2\mathbb{E}_t[\rho_{t,i}] + \frac{L\alpha^2}{2}\sum_{i=1}^{d}\mathbb{E}_t[g_{t,i}^2]\mathbb{E}_t[\rho_{t,i}^2] + \mathbb{E}_t[(\xi_{t,i}\rho_{t,i})^2]$$

$$\leq f(\boldsymbol{\theta}_t) - \alpha\bar{\rho}\sum_{i=1}^{d}\nabla[f(\boldsymbol{\theta}_t)]_i^2 + \frac{L\alpha^2}{2}\left(\sigma_n^2 + \frac{\sigma^2}{b_t} + \|\nabla f(\boldsymbol{\theta}_t)\|^2\right)$$

$$= f(\boldsymbol{\theta}_t) - \left(\alpha\bar{\rho} - \frac{L\alpha^2}{2}\right)\|\nabla f(\boldsymbol{\theta}_t)\|^2 + \frac{L\alpha^2}{2}\left(\sigma_n^2 + \frac{\sigma^2}{b_t}\right).$$

Note that $\bar{\rho} = 1 - \bar{u}$, $\mathbb{E}_t[g_{t,i}] = [\nabla f(\boldsymbol{\theta}_t)]_i$, $\mathbb{E}_t[\xi_{t,i}] = 0$, and $\mathbb{E}[(\xi_{t,i}\rho_{t,i})^2] \leq \mathbb{E}[\xi_{t,i}^2]$, since $0 \leq \rho_{t,i} \leq 1 \ \forall t, i$.

Rearranging the inequality, taking expectations on both sides, and using the telescopic sum, we can write the following:

$$\frac{2\alpha\bar{\rho} - L\alpha^2}{2} \sum_{t=1}^{T} \mathbb{E}\|\nabla f(\boldsymbol{\theta}_t)\|^2 \leq f(\boldsymbol{\theta}_1) - \mathbb{E}[f(\boldsymbol{\theta}_{T+1})] + \frac{L\alpha^2 T}{2}\left(\sigma_n^2 + \frac{\sigma^2}{b_t}\right).$$

Multiplying both sides by $\frac{2}{T(2\alpha\bar{\rho} - L\alpha^2)}$ and using the fact that $f$ is the lowest at the global minimum $\boldsymbol{\theta}^*$: $f(\boldsymbol{\theta}_{T+1}) \geq f(\boldsymbol{\theta}^*)$, we can write the following:

$$\frac{1}{T} \sum_{t=1}^{T} \mathbb{E}\|\nabla f(\boldsymbol{\theta}_t)\|^2 \leq 2\frac{f(\boldsymbol{\theta}_1) - f(\boldsymbol{\theta}^*)}{T(2\alpha\bar{\rho} - L\alpha^2)} + \frac{L\alpha^2(\sigma_n^2 b_t + \sigma^2)}{b_t(2\alpha\bar{\rho} - L\alpha^2)}.$$

Therefore, the algorithm converges to a stationary point. However, in the limit $T \to \infty$, the algorithm has to have an increasing batch size or a decreasing step size to converge, which is the same requirement for convergence of other stochastic gradient-based methods at the limit (see Zaheer et al. 2018, Ammar 2020).

## B   UTILITY PROPAGATION

The instantaneous utility measure can be used in a recursive formulation, allowing for backward propagation. We can get a recursive formula for the utility equation for connections in a neural network. This property is a result of Theorem 1.

**Theorem 1.** *If the second-order off-diagonal terms in all layers in a neural network except for the last one are zero and all higher-order derivatives are zero, the true weight utility for the weight $ij$ at the layer $l$ can be propagated using the following recursive formulation:*

$$U_{l,i,j}(Z) \doteq f_{l,i,j} + s_{l,i,j}$$

*where*

$$f_{l,i,j} \doteq \frac{\sigma'(a_{l,i})}{h_{l,i}} h_{l-1,j} W_{l,i,j} \sum_{k=1}^{|\boldsymbol{a}_{l+1}|} f_{l+1,k,i},$$

$$s_{l,i,j} \doteq \frac{1}{2} h_{l-1,j}^2 W_{l,i,j}^2 \sum_{k=1}^{|\boldsymbol{a}_{l+1}|} \left(2 s_{l+1,k,i} \frac{\sigma'(a_{l,i})^2}{h_{l,i}^2} - \frac{\sigma''(a_{l,i})}{h_{l,i}} f_{l+1,k,i}\right).$$

*Proof.* First, we start by writing the partial derivative of the loss with respect to each weight in terms of earlier partial derivatives in the next layers as follows:

$$\frac{\partial \mathcal{L}}{\partial a_{l,i}} = \sum_{k=1}^{|\boldsymbol{a}_{l+1}|} \frac{\partial \mathcal{L}}{\partial a_{l+1,k}} \frac{\partial a_{l+1,k}}{\partial h_{l,i}} \frac{\partial h_{l,i}}{\partial a_{l,i}} = \sigma'(a_{l,i}) \sum_{k=1}^{|\boldsymbol{a}_{l+1}|} \frac{\partial \mathcal{L}}{\partial a_{l+1,k}} W_{l+1,k,i}, \tag{9}$$

$$\frac{\partial \mathcal{L}}{\partial W_{l,i,j}} = \frac{\partial \mathcal{L}}{\partial a_{l,i}} \frac{\partial a_{l,i}}{\partial W_{l,i,j}} = \frac{\partial \mathcal{L}}{\partial a_{l,i}} h_{l-1,j}. \tag{10}$$

Next, we do the same with second-order partial derivatives. We use the hat notation for approximated second-order information (off-diagonal terms are dropped) as follows:

$$\widehat{\frac{\partial^2 \mathcal{L}}{\partial a_{l,i}^2}} \doteq \sum_{k=1}^{|\boldsymbol{a}_{l+1}|} \left[ \widehat{\frac{\partial^2 \mathcal{L}}{\partial a_{l+1,k}^2}} W_{l+1,k,i}^2 \sigma'(a_{l,i})^2 + \frac{\partial \mathcal{L}}{\partial a_{l+1,k}} W_{l+1,k,i} \sigma''(a_{l,i}) \right], \tag{11}$$

$$\widehat{\frac{\partial^2 \mathcal{L}}{\partial W_{l,i,j}^2}} \doteq \widehat{\frac{\partial^2 \mathcal{L}}{\partial a_{l,i}^2}} h_{l-1,j}^2. \tag{12}$$

Now, we derive the utility propagation formulation as the sum of two recursive quantities, $f_{l,ij}$ and $s_{l,ij}$. These two quantities represent the first and second-order terms in the Taylor approximation. Using Eq. 9, Eq. 10, Eq. 11, and Eq. 12, we can derive the recursive formulation as follows:

$$
\begin{aligned}
U_{l,i,j}(Z) &\doteq -\frac{\partial \mathcal{L}(\mathcal{W}, Z)}{\partial W_{l,i,j}} W_{l,i,j} + \frac{1}{2} \frac{\partial^2 \mathcal{L}(\mathcal{W}, Z)}{\partial W_{l,ij}^2} W_{l,ij}^2 \\
&\approx -\frac{\partial \mathcal{L}(\mathcal{W}, Z)}{\partial W_{l,i,j}} W_{l,i,j} + \frac{1}{2} \frac{\widehat{\partial^2 \mathcal{L}(\mathcal{W}, Z)}}{\partial W_{l,ij}^2} W_{l,ij}^2 \\
&= -\frac{\partial \mathcal{L}}{\partial a_{l,i}} h_{l-1,j} W_{l,i,j} + \frac{1}{2} \frac{\widehat{\partial^2 \mathcal{L}(\mathcal{W}, Z)}}{\partial a_{l,i,j}^2} h_{l-1,j}^2 W_{l,ij}^2 \\
&= f_{l,i,j} + s_{l,i,j}.
\end{aligned}
\tag{13}
$$

From here, we can write the first-order part $f_{l,i,j}$ and the second-order part $s_{l,i,j}$ as follows:

$$
f_{l,i,j} = -\sigma'(a_{l,i}) h_{l-1,j} W_{l,i,j} \sum_{k=1}^{|\boldsymbol{a}_{l+1}|} \left( \frac{\partial \mathcal{L}}{\partial a_{l+1,k}} W_{l+1,k,i} \right)
\tag{14}
$$

$$
s_{l,i,j} = \frac{1}{2} h_{l-1,j}^2 W_{l,i,j}^2 \sum_{k=1}^{|\boldsymbol{a}_{l+1}|} \left( \frac{\widehat{\partial^2 \mathcal{L}}}{\partial a_{l+1,k}^2} W_{l+1,k,i}^2 \sigma'(a_{l,i})^2 + \frac{\partial \mathcal{L}}{\partial a_{l+1,k}} W_{l+1,k,i} \sigma''(a_{l,i}) \right)
\tag{15}
$$

Using Eq. 14 and Eq. 15, we can write the recursive formulation for $f_{l,ij}$ and $s_{l,ij}$ as follows:

$$
\begin{aligned}
f_{l,ij} &= -\sigma'(a_{l,i}) h_{l-1,j} W_{l,i,j} \sum_{k=1}^{|\boldsymbol{a}_{l+1}|} \left( \frac{\partial \mathcal{L}}{\partial a_{l+1,k}} W_{l+1,k,i} \right) \\
&= \frac{\sigma'(a_{l,i})}{h_{l,i}} h_{l-1,j} W_{l,i,j} \sum_{k=1}^{|\boldsymbol{a}_{l+1}|} \left( -\frac{\partial \mathcal{L}}{\partial a_{l+1,k}} h_{l,i} W_{l+1,k,i} \right) \\
&= \frac{\sigma'(a_{l,i})}{h_{l,i}} h_{l-1,j} W_{l,i,j} \sum_{k=1}^{|\boldsymbol{a}_{l+1}|} f_{l+1,k,i},
\end{aligned}
\tag{16}
$$

$$
\begin{aligned}
s_{l,i,j} &= \frac{1}{2} h_{l-1,j}^2 W_{l,i,j}^2 \sum_{k=1}^{|\boldsymbol{a}_{l+1}|} \left( \frac{\widehat{\partial^2 \mathcal{L}}}{\partial a_{l+1,k}^2} W_{l+1,k,i}^2 \sigma'(a_{l,i})^2 + \frac{\partial \mathcal{L}}{\partial a_{l+1,k}} W_{l+1,k,i} \sigma''(a_{l,i}) \right) \\
&= \frac{1}{2} h_{l-1,j}^2 W_{l,i,j}^2 \sum_{k=1}^{|\boldsymbol{a}_{l+1}|} \left( \frac{\widehat{\partial^2 \mathcal{L}}}{\partial a_{l+1,k}^2} h_{l,i}^2 W_{l+1,k,i}^2 \frac{\sigma'(a_{l,i})^2}{h_{l,i}^2} - \frac{\sigma''(a_{l,i})}{h_{l,i}} f_{l+1,k,i} \right) \\
&= \frac{1}{2} h_{l-1,j}^2 W_{l,i,j}^2 \sum_{k=1}^{|\boldsymbol{a}_{l+1}|} \left( 2 s_{l+1,k,i} \frac{\sigma'(a_{l,i})^2}{h_{l,i}^2} - \frac{\sigma''(a_{l,i})}{h_{l,i}} f_{l+1,k,i} \right).
\end{aligned}
\tag{17}
$$

$\square$

## C  APPROXIMATED FEATURE UTILITY

We define the true utility of a feature as the change in the loss after the feature is removed. The utility of the feature $i$ in the $l$-th layer and sample $Z$ is given by

$$
u_{l,j}(Z) \doteq \mathcal{L}(\mathcal{W}, Z | h_{l,j} = 0) - \mathcal{L}(\mathcal{W}, Z),
\tag{18}
$$

where $h_{l,j} = 0$ denotes setting the feature output to zero (e.g., by adding a mask set to zero).

We refer to it as the *true feature utility* to distinguish it from its approximations, which are referred to as either approximated utilities or simply utilities. Note that this utility is a global measure, and

it provides a total ordering for features according to their importance. However, computing it is prohibitive since it requires additional $N_f$ forward passes, where $N_f$ is the total number of features.

Since the computation of the true feature utility is prohibitive, we aim to approximate it such that no additional forward passes are needed. We approximate the true utility of features by a second-order Taylor approximation. We expand the true utility $u_i$ around the current feature $i$ at layer $l$ and evaluate it at the value of that feature output set to zero. The quadratic approximation of $u_{l,j}(Z)$ can be written as

$$u_{l,j}(Z) = \mathcal{L}(\mathcal{W}, Z | h_{l,j} = 0) - \mathcal{L}(\mathcal{W}, Z)$$
$$\approx -\frac{\partial \mathcal{L}}{\partial h_{l,i}} h_{l,i} + \frac{1}{2} \frac{\partial^2 \mathcal{L}}{\partial h_{l,i}^2} h_{l,i}^2. \tag{19}$$

We refer to the utility measure containing the first term as the *first-order feature utility*, and the utility measure containing both terms as the *second-order feature utility*. We use the approximation by Elsayed and Mahmood (2022) that provides a Hessian diagonal approximation in linear complexity.

When we use an *origin-passing* activation function, $\sigma(0) = 0$, we can instead compute the feature utility by setting its input to zero. Thus, the feature utility can be approximated by expanding the loss around the current activation input $a_{l,i}$ as follows:

$$u_{l,j}(Z) = \mathcal{L}(\mathcal{W}, Z | a_{l,j} = 0) - \mathcal{L}(\mathcal{W}, Z)$$
$$\approx -\frac{\partial \mathcal{L}}{\partial a_{l,i}} a_{l,i} + \frac{1}{2} \frac{\partial^2 \mathcal{L}}{\partial a_{l,i}^2} a_{l,i}^2. \tag{20}$$

The gradient of the loss with respect to pre-activations or activations is not readily available using most available deep learning frameworks. To have an easily implemented algorithm, we create an equivalent rule to Eq. 18 for general activations by adding a mask on top of the activation at each layer. This mask acts as a gate on top of the activation output: $\bar{\boldsymbol{h}}_l = \boldsymbol{g}_l \circ \boldsymbol{h}_l$. Note that the weights of such gates are set to ones and never change throughout learning. The quadratic approximation of $u_{l,j}(Z)$ can be written as

$$u_{l,j}(Z) = \mathcal{L}(\mathcal{W}, Z | g_{l,j} = 0) - \mathcal{L}(\mathcal{W}, Z)$$
$$\approx \mathcal{L}(\mathcal{W}, Z) + \frac{\partial \mathcal{L}}{\partial g_{l,i}}(0 - g_{l,j}) + \frac{1}{2} \frac{\partial^2 \mathcal{L}}{\partial g_{l,i}^2}(0 - g_{l,j})^2 - \mathcal{L}(\mathcal{W}, Z)$$
$$= -\frac{\partial \mathcal{L}}{\partial g_{l,i}} + \frac{1}{2} \frac{\partial^2 \mathcal{L}}{\partial g_{l,i}^2}. \tag{21}$$

For feature-wise UPGD, the global scaled utility is given by $\bar{u}_{l,j} = \phi(u_{l,j}/\eta)$, where $\eta$ is the maximum utility of the features and $\phi$ is the scaling function, for which we use sigmoid, with its corresponding Algorithm 3, whereas UPGD with the local scaled utility is given by $\bar{u}_{l,j} = \phi\left(u_{l,j}/\sqrt{\sum_j u_{l,j}^2}\right)$ in Algorithm 4.

The approximated feature utility can be computed using the approximated weight utility, which gives rise to the *conservation of utility* property. We show this relationship in Appendix D.

## D    FEATURE UTILITY APPROXIMATION USING WEIGHT UTILITY

Instead of deriving feature utility directly, we derive the utility of a feature based on setting its output weights to zero. Equivalently to Eq. 18, the utility of a feature $i$ at layer $l$ is given by

$$u_{l,i}(Z) = \mathcal{L}(\mathcal{W}_{\neg[l,i]}, Z) - \mathcal{L}(\mathcal{W}, Z), \tag{22}$$

where $\mathcal{W}$ is the set of all weights, $\mathcal{L}(\mathcal{W}, Z)$ is the sample loss of a network parameterized by $\mathcal{W}$ on sample $Z$, and $\mathcal{W}_{\neg[l,i]}$ is the same as $\mathcal{W}$ except the weight $\boldsymbol{W}_{l+1,i,j}$ is set to 0 for all values of $i$.

Note that the second-order Talyor's approximation of this utility depends on the off-diagonal elements of the Hessian matrix at each layer, since more than one weight is removed at once. For our analysis, we drop these elements and derive our feature utility. We expand the difference around the

current output weights of the feature $i$ at layer $l$ and evaluate it by setting the weights to zero. The quadratic approximation of $u_{l,i}(Z)$ can be written as

$$
\begin{aligned}
u_{l,j}(Z) &= \mathcal{L}(\mathcal{W}_{\neg[l,i]}, Z) - \mathcal{L}(\mathcal{W}, Z) \\
&= \sum_{i=1}^{|\boldsymbol{a}_{l+1}|} \left( -\frac{\partial \mathcal{L}}{\partial W_{l+1,i,j}} W_{l+1,i,j} + \frac{1}{2} \frac{\partial^2 \mathcal{L}}{\partial W_{l+1,ij}^2} W_{l+1,i,j}^2 \right) \\
&\quad + \sum_{i=1}^{|\boldsymbol{a}_{l+1}|} \left( 2 \sum_{j \neq i} \frac{\partial^2 \mathcal{L}}{\partial W_{l+1,i,j} \partial W_{l+1,i,k}} W_{l+1,i,j} W_{l+1,i,k} \right) \\
&\approx \sum_{i=1}^{|\boldsymbol{a}_{l+1}|} \left( -\frac{\partial \mathcal{L}}{\partial W_{l+1,ij}} W_{l+1,ij} + \frac{1}{2} \frac{\partial^2 \mathcal{L}}{\partial W_{l+1,ij}^2} W_{l+1,ij}^2 \right) \\
&= \sum_{i=1}^{|\boldsymbol{a}_{l+1}|} U_{l+1,i,j}.
\end{aligned}
\tag{23}
$$

Alternatively, we can derive the utility of feature $i$ at layer $l$ by dropping the input weights when the activation function passes through the origin (zero input leads to zero output). This gives rise to the property of *conservation of utility* shown by Therom 2.

**Theorem 2.** *If the second-order off-diagonals in all layers in a neural network except for the last one are zero, all higher-order derivatives are zero, and an origin-passing activation function is used, the sum of output-weight utilities to a feature equals the sum of its input-weight utilities.*

$$
\sum_{i=1}^{|\boldsymbol{a}_{l+1}|} U_{l+1,i,j} = \sum_{i=1}^{|\boldsymbol{a}_l|} U_{l,j,i}.
$$

*Proof.* From Eq. 23, we can write the utility of the feature $j$ at layer $l$ by dropping the output weights. The sample feature utility of is given by

$$
u_{l,j}(Z) = \sum_{i=1}^{|\boldsymbol{a}_{l+1}|} U_{l+1,i,j}(Z).
$$

Similarly, we can write the utility of the feature $j$ at layer $l$ by dropping the input weights when the activation function passes through the origin (zero input leads to zero output) as follows:

$$
u_{l,j}(Z) = \sum_{i=1}^{|\boldsymbol{a}_l|} U_{l,j,i}(Z).
$$

Therefore, we can write the following equality:

$$
\sum_{i=1}^{|\boldsymbol{a}_{l+1}|} U_{l+1,i,j} = \sum_{i=1}^{|\boldsymbol{a}_l|} U_{l,j,i}.
$$

$\square$

This theorem shows the property of the conservation of instantaneous utility. The sum of utilities of the outgoing weights to a feature equals the sum of utilities of the incoming weights to the same feature when origin-passing activation functions are used and the off-diagonal elements are dropped. This conservation law resembles the one introduced by Tanaka et al. (2020).

# E  WEIGHT-WISE UPGD, FEATURE-WISE UPGD, AND ADAPTIVE UPGD

---

**Algorithm 2** Weight-wise UPGD with local utility

---

Given a stream of data $\mathcal{D}$ and a neural network $f$ with weights $\{\boldsymbol{W}_1, ..., \boldsymbol{W}_L\}$.
Initialize step size $\alpha$, utility decay rate $\beta$, and noise standard deviation $\sigma$.
Initialize $\{\boldsymbol{W}_1, ..., \boldsymbol{W}_L\}$.
Initialize $\boldsymbol{U}_l, \forall l$ and time step $t$ to zero.
**for** $(\boldsymbol{x}, \boldsymbol{y})$ in $\mathcal{D}$ **do**
 $t \leftarrow t + 1$
 **for** $l$ in $\{L, L-1, ..., 1\}$ **do**
  $\boldsymbol{F}_l, \boldsymbol{S}_l \leftarrow \texttt{GetDerivatives}(f, \boldsymbol{x}, \boldsymbol{y}, l)$
  $\boldsymbol{M}_l \leftarrow {}^1\!/_2 \boldsymbol{S}_l \circ \boldsymbol{W}_l^2 - \boldsymbol{F}_l \circ \boldsymbol{W}$
  $\boldsymbol{U}_l \leftarrow \beta \boldsymbol{U}_l + (1 - \beta) \boldsymbol{M}_l$
  $\hat{\boldsymbol{U}}_l \leftarrow \boldsymbol{U}_l / (1 - \beta^t)$
  Sample $\boldsymbol{\xi}$ elements from $\mathcal{N}(0, \sigma^2)$
  $\bar{\boldsymbol{U}}_l \leftarrow \phi(\boldsymbol{D}\hat{\boldsymbol{U}}_l)$      $\triangleright D_{ii} = 1/\|\boldsymbol{U}_{l,i,:}\|$ and $D_{ij} = 0, \forall i \neq j$
  $\boldsymbol{W}_l \leftarrow \boldsymbol{W}_l - \alpha(\boldsymbol{F}_l + \boldsymbol{\xi}) \circ (1 - \bar{\boldsymbol{U}}_l)$

---

| **Algorithm 3** Feature-wise UPGD w/ global utility | **Algorithm 4** Feature-wise UPGD w/ local utility |
|---|---|
| Given a stream of data $\mathcal{D}$ and a neural network $f$ with weights $\{\boldsymbol{W}_1, ..., \boldsymbol{W}_L\}$. 
 Initialize step size $\alpha$, utility decay rate $\beta$, and noise standard deviation $\sigma$. 
 Initialize $\{\boldsymbol{W}_1, ..., \boldsymbol{W}_L\}$ randomly. 
 Initialize $\{\boldsymbol{g}_1, ..., \boldsymbol{g}_{L-1}\}$ to ones. 
 Initialize $\boldsymbol{u}_l, \forall l$ and time step $t$ to zero. 
 $t \leftarrow t + 1$ 
 **for** $(\boldsymbol{x}, \boldsymbol{y})$ in $\mathcal{D}$ **do** 
  $t \leftarrow t + 1$ 
  **for** $l$ in $\{L-1, ..., 1\}$ **do** 
   $\eta \leftarrow -\infty$ 
   $\boldsymbol{F}_{l,\_} \leftarrow \texttt{GetDerivatives}(f, \boldsymbol{x}, \boldsymbol{y}, l)$ 
   $\boldsymbol{f}_l, \boldsymbol{s}_l \leftarrow \texttt{GateDerivatives}(f, \boldsymbol{x}, \boldsymbol{y}, l)$ 
   $\boldsymbol{m}_l \leftarrow {}^1\!/_2 \boldsymbol{s}_l - \boldsymbol{f}_l$ 
   $\boldsymbol{u}_l \leftarrow \beta \boldsymbol{u}_l + (1 - \beta) \boldsymbol{m}_l$ 
   $\hat{\boldsymbol{u}}_l \leftarrow \boldsymbol{u}_l / (1 - \beta^t)$ 
   **if** $\eta < \max(\hat{\boldsymbol{u}}_l)$ **then** $\eta \leftarrow \max(\hat{\boldsymbol{u}}_l)$ 
  **for** $l$ in $\{L, L-1, ..., 1\}$ **do** 
   **if** $l = L$ **then** 
    $\boldsymbol{W}_L \leftarrow \boldsymbol{W}_L - \alpha \boldsymbol{F}_L$ 
   **else** 
    $\bar{\boldsymbol{u}}_l \leftarrow \phi(\hat{\boldsymbol{u}}_l / \eta)$ 
    Sample $\boldsymbol{\xi}$ elements from $\mathcal{N}(0, \sigma^2)$ 
    $\boldsymbol{W}_l \leftarrow \boldsymbol{W}_l - \alpha(\boldsymbol{F}_l + \boldsymbol{\xi}) \circ (1 - \boldsymbol{1}\bar{\boldsymbol{u}}_l^\top)$ | Given a stream of data $\mathcal{D}$ and a neural network $f$ with weights $\{\boldsymbol{W}_1, ..., \boldsymbol{W}_L\}$. 
 Initialize step size $\alpha$, utility decay rate $\beta$, and noise standard deviation $\sigma$. 
 Initialize $\{\boldsymbol{W}_1, ..., \boldsymbol{W}_L\}$ randomly. 
 Initialize $\{\boldsymbol{g}_1, ..., \boldsymbol{g}_{L-1}\}$ to ones. 
 Initialize $\boldsymbol{u}_l, \forall l$ and time step $t$ to zero. 
 $t \leftarrow t + 1$ 
 **for** $(\boldsymbol{x}, \boldsymbol{y})$ in $\mathcal{D}$ **do** 
  $t \leftarrow t + 1$ 
  **for** $l$ in $\{L, L-1, ..., 1\}$ **do** 
   $\boldsymbol{F}_{l,\_} \leftarrow \texttt{GetDerivatives}(f, \boldsymbol{x}, \boldsymbol{y}, l)$ 
   **if** $l = L$ **then** 
    $\boldsymbol{W}_L \leftarrow \boldsymbol{W}_L - \alpha \boldsymbol{F}_L$ 
   **else** 
    $\boldsymbol{f}_l, \boldsymbol{s}_l \leftarrow \texttt{GateDerivatives}(f, \boldsymbol{x}, \boldsymbol{y}, l)$ 
    $\boldsymbol{m}_l \leftarrow {}^1\!/_2 \boldsymbol{s}_l - \boldsymbol{f}_l$ 
    $\boldsymbol{u}_l \leftarrow \beta \boldsymbol{u}_l + (1 - \beta) \boldsymbol{m}_l$ 
    $\hat{\boldsymbol{u}}_l \leftarrow \boldsymbol{u}_l / (1 - \beta^t)$ 
    $\bar{\boldsymbol{u}}_l \leftarrow \phi(\hat{\boldsymbol{u}}_l / \|\hat{\boldsymbol{u}}_l\|)$ 
    Sample $\boldsymbol{\xi}$ elements from $\mathcal{N}(0, \sigma^2)$ 
    $\boldsymbol{W}_l \leftarrow \boldsymbol{W}_l - \alpha(\boldsymbol{F}_l + \boldsymbol{\xi}) \circ (1 - \boldsymbol{1}\bar{\boldsymbol{u}}_l^\top)$ |

---

# F  THE GETDERIVATIVES AND GATEDERIVATIVES FUNCTIONS

Here, we describe GetDerivatives used in UPGD and GateDerivatives used in the feature-wise variation of UPGD. Each function takes four arguments: the neural network $f$, the input $\boldsymbol{x}$, the target $\boldsymbol{y}$, and the layer number $l$. The GetDerivatives function returns the gradient of the loss with respect to the weight matrix $\boldsymbol{W}_l$ given by $\boldsymbol{F}_l$ and the approximated second-order information of the loss with respect to the matrix $\boldsymbol{W}_l$ given by $\boldsymbol{S}_l$. The GateDerivatives function is similar, which returns the gradient of the loss with respect to the gate $\boldsymbol{g}_l$ given by $\boldsymbol{f}_l$ and the approximated second-order information of the loss with respect to the gate $\boldsymbol{g}_l$ given by $\boldsymbol{s}_l$. The matrix $\boldsymbol{S}_l$ and the vector $\boldsymbol{s}_l$ store the diagonal approximation of $\text{Diag}(\nabla^2_{\boldsymbol{W}_l}\mathcal{L})$ and $\text{Diag}(\nabla^2_{\boldsymbol{g}_l}\mathcal{L})$ reshaped to be of the same size as $\boldsymbol{W}_l$ and $\boldsymbol{g}_l$, respectively. While different methods can be used to get the derivatives, UPGD uses HesScale for it, which is given by Algorithm 6.

---

**Algorithm 5** AdaUPGD: Adaptive Utility-based Gradient Descent

---

Given a stream of data $\mathcal{D}$, a network $f$ with weights $\{\boldsymbol{W}_1, ..., \boldsymbol{W}_L\}$.
Initialize utility decay rate $\beta_u$, momentum decay rate $\beta_1$, and RMSprop decay rate $\beta_2$, where $\beta_u, \beta_1, \beta_2 \in [0, 1)$ and set a small number $\epsilon$ for numerical stability (e.g., $10^{-8}$).
Initialize step size $\alpha$ and noise standard deviation $\sigma$.
Initialize $\{\boldsymbol{W}_1, ..., \boldsymbol{W}_L\}$.
Initialize $\boldsymbol{U}_l, \boldsymbol{M}_l, \boldsymbol{V}_l, \forall l$ and time step $t$ to zero.
**for** $(\boldsymbol{x}, \boldsymbol{y})$ in $\mathcal{D}$ **do**
  $t \leftarrow t + 1$
  **for** $l$ in $\{L, L-1, ..., 1\}$ **do**
    $\eta \leftarrow -\infty$
    $\boldsymbol{F}_l, \boldsymbol{S}_l \leftarrow \texttt{GetDerivatives}(f, \boldsymbol{x}, \boldsymbol{y}, l)$
    $\boldsymbol{R}_l \leftarrow {}^1\!/_2 \boldsymbol{S}_l \circ \boldsymbol{W}_l^2 - \boldsymbol{F}_l \circ \boldsymbol{W}_l$
    $\boldsymbol{U}_l \leftarrow \beta_u \boldsymbol{U}_l + (1 - \beta_u) \boldsymbol{R}_l$
    $\boldsymbol{M}_l \leftarrow \beta_1 \boldsymbol{M}_l + (1 - \beta_1) \boldsymbol{F}_l$
    $\boldsymbol{V}_l \leftarrow \beta_2 \boldsymbol{V}_l + (1 - \beta_2) \boldsymbol{F}_l^{\circ 2}$
    $\hat{\boldsymbol{U}}_l \leftarrow \boldsymbol{U}_l / (1 - \beta_u^t)$
    $\hat{\boldsymbol{M}}_l \leftarrow \boldsymbol{M}_l / (1 - \beta_1^t)$
    $\hat{\boldsymbol{V}}_l \leftarrow \boldsymbol{V}_l / (1 - \beta_2^t)$
    **if** $\eta < \max(\hat{\boldsymbol{U}}_l)$ **then**
      $\eta \leftarrow \max(\hat{\boldsymbol{U}}_l)$
  **for** $l$ in $\{L, L-1, ..., 1\}$ **do**
    Sample $\boldsymbol{\xi}$ elements from $\mathcal{N}(0, \sigma^2)$
    $\bar{\boldsymbol{U}}_l \leftarrow \phi(\hat{\boldsymbol{U}}_l / \eta)$
    $\boldsymbol{W}_l \leftarrow \boldsymbol{W}_l - \alpha \left( \hat{\boldsymbol{M}}_l \oslash \left( \hat{\boldsymbol{V}}^{\circ \frac{1}{2}} + \epsilon \right) + \boldsymbol{\xi} \right) \circ (1 - \bar{\boldsymbol{U}}_l)$

---

**Algorithm 6** The HesScale Algorithm in Classification (Elsayed & Mahmood 2022)

---

**Require:** Neural network $f$ and a layer number $l$
**Require:** $\widehat{\frac{\partial \mathcal{L}}{\partial \boldsymbol{a}_{l+1}}}$ and $\widehat{\frac{\partial^2 \mathcal{L}}{\partial \boldsymbol{a}_{l+1}^2}}$, unless $l = L$
**Require:** Input-output pair $(\boldsymbol{x}, y)$.
Compute preference vector $\boldsymbol{a}_L \leftarrow f(\boldsymbol{x})$ and target one-hot-encoded vector $\boldsymbol{p} \leftarrow \texttt{onehot}(y)$.
Compute the predicted probability vector $\boldsymbol{q} \leftarrow \boldsymbol{\sigma}(\boldsymbol{a}_L)$ and Compute the loss $\mathcal{L}(\boldsymbol{p}, \boldsymbol{q})$.
**if** $l = L$ **then**
  Compute $\frac{\partial \mathcal{L}}{\partial \boldsymbol{a}_L} \leftarrow \boldsymbol{q} - \boldsymbol{p}$
  Compute $\frac{\partial \mathcal{L}}{\partial \boldsymbol{W}_L}$ using Eq. 10
  $\widehat{\frac{\partial^2 \mathcal{L}}{\partial \boldsymbol{a}_L^2}} \leftarrow \boldsymbol{q} - \boldsymbol{q} \circ \boldsymbol{q}$
  Compute $\widehat{\frac{\partial^2 \mathcal{L}}{\partial \boldsymbol{W}_L^2}}$ using Eq. 12
**else**
  Compute $\frac{\partial \mathcal{L}}{\partial \boldsymbol{a}_l}$ and $\partial \mathcal{L} / \partial \boldsymbol{W}_l$ using Eq. 9 and Eq. 10
  Compute $\widehat{\frac{\partial^2 \mathcal{L}}{\partial \boldsymbol{a}_l^2}}$ and $\widehat{\frac{\partial^2 \mathcal{L}}{\partial \boldsymbol{W}_l^2}}$ using Eq. 11 and Eq. 12
**Return** $\frac{\partial \mathcal{L}}{\partial \boldsymbol{W}_l}, \widehat{\frac{\partial^2 \mathcal{L}}{\partial \boldsymbol{W}_l^2}}, \frac{\partial \mathcal{L}}{\partial \boldsymbol{a}_l}$, and $\widehat{\frac{\partial^2 \mathcal{L}}{\partial \boldsymbol{a}_l^2}}$

---

# G UPGD ON STATIONARY MNIST

We use the MNIST dataset to assess the performance of UPGD under stationarity. A desirable property of continual learning systems is that they should not asymptotically impose any extra performance reduction, which can be studied in a stationary task such as MNIST. We report the results in Fig. 9. We notice that UPGD improves performance over SGD. Each point in the stationary MNIST plots represents an average accuracy over a non-overlapping window of 10000 samples.

The learners use a network of $300 \times 150$ units with ReLU activations. We used the hyperparameter search space shown in Table 1. The utility traces are computed using exponential moving averages given by $\tilde{U}_t = \beta_u \tilde{U}_{t-1} + (1-\beta_u)U_t$, where $\tilde{U}_t$ is the utility trace at time $t$ and $U_t$ is the instantaneous utility at time $t$.

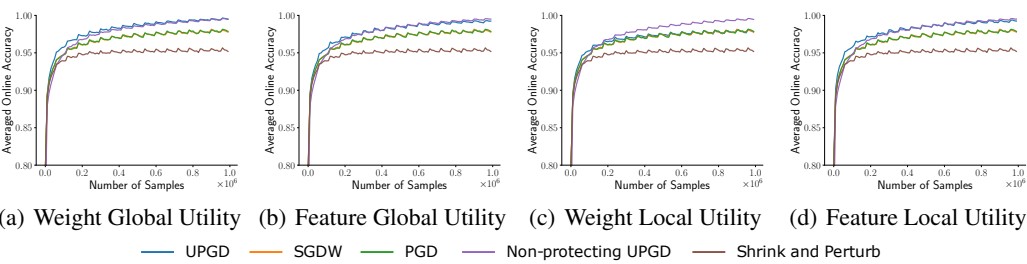

(a) Weight Global Utility   (b) Feature Global Utility   (c) Weight Local Utility   (d) Feature Local Utility

Figure 9: Performance of Utility-based Perturbed Gradient Descent with first-order approximated utilities on stationary MNIST. The results are averaged over 20 independent runs.

## H   UPGD ON NON-STATIONARY TOY REGRESSION PROBLEM

We study UPGD's effectiveness in mitigating catastrophic forgetting and loss of plasticity in a simple toy regression problem that is easy to analyze and understand. The target at time $t$ is given by $y_t = \frac{a}{|\mathcal{S}|} \sum_{i \in \mathcal{S}} x_{t,i}$, where $x_{t,i}$ is the $i$-th entry of input vector at time $t$, $\mathcal{S}$ is a set of some input indices, and $a \in \{-1, 1\}$. The inputs are sampled from $\mathcal{N}(0, 1)$.

In this problem, the task is to calculate the average of two inputs or its negative out of 16 inputs. We introduce non-stationarity using two ways: changing the multiplier $a$ or changing the input-index set $\mathcal{S}$. The learner is required to match the targets by minimizing the online squared error. The learner uses a multi-layer ($300 \times 150$) linear network, where the activation used is the identity activation ($\sigma(\boldsymbol{x}) = \boldsymbol{x}$). We use linear activations to see if catastrophic forgetting and loss of plasticity may occur even in such simple networks.

The first variation of the problem focuses solely on loss of plasticity. We can study plasticity when the learner is presented with sequential tasks requiring little transfer between them. Here, $|\mathcal{S}| = 2$ and the input indices change every 200 time steps by a shift of two. For instance, if the first task has $\mathcal{S} = \{1, 2\}$, the next would be $\{3, 4\}$ and so on. Since the tasks share little similarity between them, we expect the continual learners to learn as quickly as possible by discarding old features when needed to maintain their plasticity. We compare UPGD against SGD, PGD, S&P, and Non-protecting UPGD. We also use a baseline with one linear layer mapping the input to the output.

The second variation of the problem focuses on catastrophic forgetting. Here, the sign of the target sum is flipped every 200 time steps by changing $a$ from 1 to $-1$ and vice versa. Since the two tasks share high similarities, we expect continual learners to initially learn some features during the first 200 steps. Then, after the target sign flip, we expect the learner to change the sign of only the output weights and keep the features intact since the previously learned features are fully useful. The frequency of changing $a$ is high to penalize learners for re-learning features from scratch.

For the input-index changing problem, Fig. 10 shows that the performance of SGD degrades with changing targets even when using linear neural networks, indicating SGD loses plasticity over time. Each point in the toy problem figures represents an average squared error of 20 tasks. The results are averaged over 20 independent runs. Empirically, we found that the outgoing weights of the last layer get smaller, hindering the ability to change the features' input weights. S&P outperforms the linear-layer baseline. However, PGD and Non-protecting UPGD perform better than S&P, indicating that weight decay is not helpful in this problem, and it is better to just inject noise without shrinking the parameters. UPGD cannot only maintain its plasticity but also improve its performance rapidly with changing targets compared to other methods.

For the output-sign changing problem, Fig. 11 shows that the performance of SGD degrades with changing targets, indicating that it does not utilize learned features and re-learn them every time

the targets change. S&P, Non-protecting UPGD, and PGD do not lose plasticity over time but perform worse than UPGD, indicating that they are ineffective in protecting useful weights. UPGD is likely protecting and utilizing useful weights in subsequent tasks. Moreover, the performance keeps improving with changing targets compared to the other methods.

In both variations, we use utility traces (e.g., using exponential moving average) instead of instantaneous utility, as we empirically found utility traces to perform better. We found that the second-order approximated utility improves performance over the first-order approximated utility in these problems. A hyperparameter search, given in Table 1, was conducted. The best-performing hyperparameter set was used for plotting (see Table 3 for weight-wise UPGD alongside other algorithms).

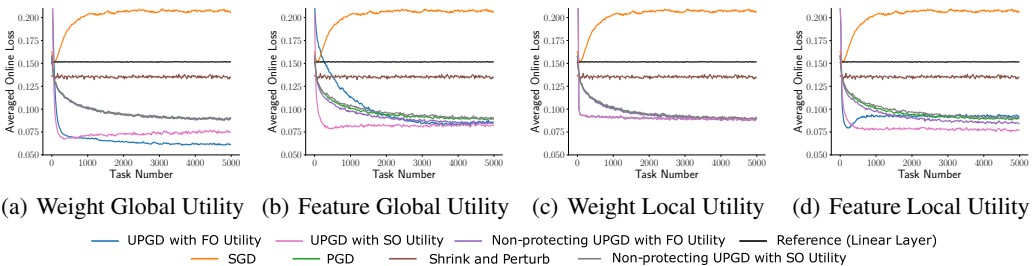

(a) Weight Global Utility    (b) Feature Global Utility    (c) Weight Local Utility    (d) Feature Local Utility

Figure 10: Performance of UPGD on the toy problem with a changing input-index set against SGD, PGD, and S&P. First-order and second-order approximated utilities are used.

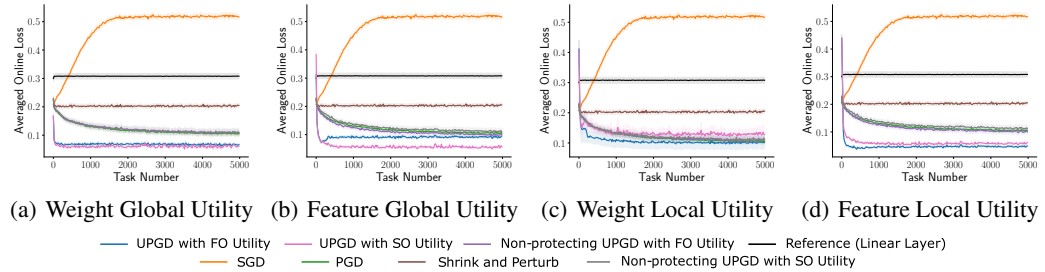

(a) Weight Global Utility    (b) Feature Global Utility    (c) Weight Local Utility    (d) Feature Local Utility

Figure 11: Performance of UPGD on the toy problem with changing output sign against SGD, PGD, and S&P. First-order and second-order approximated utilities are used.

## I EXPERIMENTAL DETAILS

In each experiment, the utility traces are computed using exponential moving averages given by $\tilde{U}_t = \beta_u \tilde{U}_{t-1} + (1 - \beta_u) U_t$, where $\tilde{U}_t$ is the utility trace at time $t$ and $U_t$ is the instantaneous utility at time $t$. Each learner is trained for 1 million time steps, except for the experiment in Fig. 1(b) and its closer look in Fig. 1(c), which we run for 15 million time steps.

### I.1 EXPERIMENTS IN FIG. 1

In the problem of Label-permuted EMNIST, the labels are permuted every 2500 sample. For the offline variation, we form a held-out set comprised of 250 samples of the test set of EMNIST. The performance is measured based on the learner's prediction on this held-out set after the end of each task in an offline manner. We permute the labels of the held-out set with the same permutation in the task presented to the learner to have consistent evaluation. We observe the same phenomenon indicated by Lesort et al. (2023), in which the performance of SGD decreases on the first few tasks, matching the vast majority of works on catastrophic forgetting. However, when scaling the number of tasks, SGD recovers and maintains its performance. The online variation uses the same experimental details in Section 4.4. The hyper-parameters used are given in Table 3.

In the problem of input-permuted MNIST, the inputs are permuted every 60000 sample. In Fig. 1(c), we show the online accuracy in 4 tasks where each point represents an average of over 600 samples.

The step-size $\alpha$ used for Adam is $0.0001$ using the default $\beta_1 = 0.9$, $\beta_2 = 0.9999$, and $\epsilon = 10^{-8}$. We used for UPGD-W a step size $\alpha$ of $0.01$, a utility trace decay factor $\beta_u$ of $0.999$, a weight decay $\lambda$ of $0.001$, and a noise standard deviation $\sigma$ of $0.01$.

## I.2 Input-permuted MNIST and Label-permuted EMNIST/mini-ImageNet

Each point in the Input-permuted MNIST and Label-permuted EMNIST figures represents an average accuracy of one task. The learners use a network of two hidden layers containing 300 and 150 units with ReLU activations, respectively. The results are averaged over 20 independent runs, and the shaded area represents the standard error.

## I.3 Label-permuted CIFAR-10

The learners use a network with two convolutional layers with max-pooling followed by two fully connected layers with ReLU activations. The first convolutional layer uses a kernel of 5 and outputs 6 filters, whereas the second convolutional layer uses a kernel of 5 and outputs 16 filters. The max-pooling layers use a kernel of 2. The data is flattened after the second max-pooling layer and fed to a fully connected network with two hidden layers containing 120 and 84 units, respectively. The results are averaged over 10 independent runs, and the shaded area represents the standard error.

## I.4 S-SI and S-RWalk Details

We compute weight importance $\Omega_{l,i,j}$ in S-SI by first maintaining a trace of the gradient multiplied by the weight change from its previous value (with a decay factor $\beta_I$). We then maintain a trace of the weight change from its previous value (with a decay factor $\beta_I$). The weight importance for S-SI is defined as the ratio of the first trace and the squared value of the second trace as follows:

$$\Omega_{l,i,j} = \frac{\tilde{\omega}_{l,i,j}}{(\tilde{\Delta}_{l,i,j})^2 + \epsilon},$$

where $\tilde{\omega}_{l,i,j}$ is a moving average of the gradient $g_{l,i,j}$ multiplied by the weight change from its previous weight $w^{-}_{l,i,k}$ as: $g_{l,i,j}(w^{-}_{l,i,k} - w_{l,i,k})$, $\tilde{\Delta}_{l,i,j}$ is a moving average of the weight change from its previous weight as $(w^{-}_{l,i,k} - w_{l,i,k})$, and $\epsilon$ is a small number for numerical stability which we set to $10^{-3}$. Note that the update rule uses a trace of weights with a decay rate $\beta_w$.

On the other hand, the weight importance estimation in S-RWalk is computed as:

$$s_{l,i,j} = \frac{\tilde{\omega}_{l,i,j}}{\frac{1}{2}\tilde{F}_{l,i,j}(w_{l,i,k} - w^{-}_{l,i,k})^2 + \epsilon},$$
$$\Omega_{l,i,j} = \tilde{s}_{l,i,j} + F_{l,i,j},$$

where $\tilde{F}_{l,i,j}$ is a moving average of past squared gradients (with a decay factor $\beta_I$). Note that $\tilde{s}_{l,i,j}$ is estimated with a fast-moving average with a decay factor of $0.5$ based on $s_{l,i,j}$.

## I.5 Hyperparameter Search Space

In this section, we present the hyperparameter search space we conduct for each method in each problem in Table 1. Our grid search is quite exhaustive and can reach up to 100 configurations per algorithm for a single run. Table 3 and Table 2 show the best hyperparameter configuration for each method on each problem.

## I.6 Policy Collapse Experiment

We evaluate AdaUPGD using various MuJoCo (Todorov et al. 2012) environments in comparison to Adam. We use the CleanRL (Huang et al. 2022) implementation for PPO with its default best hyperparameters. For AdaUPGD, we used the same values of $\beta_1$, $\beta_2$, and $\epsilon$ used in Adam, which are $0.9$, $0.999$, and $10^{-5}$, respectively. We used noise with a standard deviation of $0.001$ and a utility trace with a decay rate of $0.999$ in all environments except for *Humanoid-v4* and *HumanoidStandup-v4*, in which we use a utility trace with a decay rate of $0.9$.

Table 1: Hyperparameter search spaces. N-UPGD is short for non-protecting UPGD.

| Problem | | Space | Method |
|---|---|---|---|
| **Toy Problems** **Stationary MNIST** | $\alpha$ | $\{0.1, 0.01, 0.001, 0.0001\}$ | All |
| | $\beta_u$ | $\{0.0, 0.9, 0.99, 0.999\}$ | UPGD, N-UPGD |
| | $\sigma$ | $\{0.001, 0.01, 0.1, 1.0\}$ | S&P, UPGD, N-UPGD |
| | $\lambda$ | $\{0.1, 0.01, 0.001, 0.0001\}$ | S&P |
| **Input-permuted MNIST** **Label-permuted EMNIST** **Label-permuted CIFAR-10** **Label-permuted *mini*-ImageNet** | $\alpha$ | $\{0.1, 0.01, 0.001, 0.0001\}$ | All |
| | $\beta_u$ | $\{0.9, 0.99, 0.999, 0.9999\}$ | UPGD, N-UPGD |
| | $\sigma$ | $\{0.001, 0.01, 0.1, 1.0\}$ | S&P, UPGD, N-UPGD |
| | $\lambda$ | $\{0.0, 0.1, 0.01, 0.001, 0.0001\}$ | S&P, UPGD, N-UPGD |
| | $\beta_1$ | $\{0.0, 0.9\}$ | Adam |
| | $\beta_2$ | $\{0.99, 0.999, 0.9999\}$ | Adam |
| | $\epsilon$ | $\{10^{-4}, 10^{-8}, 10^{-16}\}$ | Adam |
| | $\kappa$ | $\{100, 10.0, 1.0, 0.1, 0.01\}$ | S-EWC, S-SI, S-MAS, S-RWalk |
| | $\beta_I$ | $\{0.9, 0.99, 0.999, 0.9999\}$ | S-EWC, S-SI, S-MAS, S-RWalk |
| | $\beta_w$ | $\{0.9, 0.99, 0.999\}$ | S-EWC, S-SI, S-MAS, S-RWalk |

Table 2: Best hyperparameter set of each method on Label-permuted CIFAR10.

| Problem | Method | Best Set |
|---|---|---|
| **Label-permuted CIFAR-10** | SGDW | $\alpha = 0.01, \lambda = 0.001$ |
| | S&P | $\alpha = 0.01, \sigma = 0.01, \lambda = 0.001$ |
| | PGD | $\alpha = 0.001, \sigma = 0.01$ |
| | AdamW | $\alpha = 0.001, \beta_1 = 0.0, \beta_2 = 0.9999, \epsilon = 10^{-8}, \lambda = 0.01$ |
| | S-EWC | $\alpha = 0.01, \beta_I = 0.9999, \beta_w = 0.999, \kappa = 10.0$ |
| | S-MAS | $\alpha = 0.01, \beta_I = 0.9999, \beta_w = 0.999, \kappa = 10.0$ |
| | S-SI | $\alpha = 0.001, \beta_I = 0.99, \beta_w = 0.99, \kappa = 0.01$ |
| | S-RWalk | $\alpha = 0.001, \beta_I = 0.9, \beta_w = 0.999, \kappa = 10.0$ |
| | UPGD-W | $\alpha = 0.01, \sigma = 0.001, \beta_u = 0.999, \lambda = 0.0$ |
| | N-UPGD-W | $\alpha = 0.01, \sigma = 0.01, \beta_u = 0.99, \lambda = 0.001$ |

Table 3: Best hyperparameter set of each method in the toy problems, Input-permuted MNIST, Label-permuted EMNIST, and Label-permuted *mini*-ImageNet.

| Problem | Method | Best Set |
|---|---|---|
| **Toy Problem 1** | SGD | $\alpha = 0.01$ |
| | PGD | $\alpha = 0.01, \sigma = 0.1$ |
| | S&P | $\alpha = 0.01, \sigma = 1.0, \lambda = 0.1$ |
| | UPGD | $\alpha = 0.01, \sigma = 0.0001, \beta_u = 0.9$ |
| | N-UPGD | $\alpha = 0.01, \sigma = 0.1, \beta_u = 0.999$ |
| | Reference | $\alpha = 0.01$ |
| **Toy Problem 2** | SGD | $\alpha = 0.01$ |
| | PGD | $\alpha = 0.01, \sigma = 0.1$ |
| | S&P | $\alpha = 0.01, \sigma = 1.0, \lambda = 0.1$ |
| | UPGD | $\alpha = 0.01, \sigma = 0.1, \beta_u = 0.9$ |
| | N-UPGD | $\alpha = 0.01, \sigma = 0.1, \beta_u = 0.999$ |
| | Reference | $\alpha = 0.01$ |
| **Input-permuted MNIST** | SGDW | $\alpha = 0.001, \lambda = 0.001$ |
| | S&P | $\alpha = 0.001, \sigma = 0.1, \lambda = 0.01$ |
| | PGD | $\alpha = 0.001, \sigma = 0.1$ |
| | AdamW | $\alpha = 0.0001, \beta_1 = 0.0, \beta_2 = 0.99, \epsilon = 10^{-8}, \lambda = 0.0$ |
| | S-EWC | $\alpha = 0.001, \beta_I = 0.9999, \beta_w = 0.99, \kappa = 0.001$ |
| | S-MAS | $\alpha = 0.001, \beta_I = 0.9999, \beta_w = 0.999, \kappa = 0.1$ |
| | S-SI | $\alpha = 0.001, \beta_I = 0.9999, \beta_w = 0.999, \kappa = 0.1$ |
| | S-RWalk | $\alpha = 0.001, \beta_I = 0.99, \beta_w = 0.999, \kappa = 10.0$ |
| | UPGD-W | $\alpha = 0.01, \sigma = 0.1, \beta_u = 0.9999, \lambda = 0.01$ |
| | N-UPGD-W | $\alpha = 0.001, \sigma = 0.1, \beta_u = 0.9, \lambda = 0.01$ |
| **Label-permuted EMNIST** | SGDW | $\alpha = 0.01, \lambda = 0.0001$ |
| | S&P | $\alpha = 0.01, \sigma = 0.01, \lambda = 0.001$ |
| | PGD | $\alpha = 0.01, \sigma = 0.01$ |
| | AdamW | $\alpha = 0.0001, \beta_1 = 0.0, \beta_2 = 0.9999, \epsilon = 10^{-8}, \lambda = 0.1$ |
| | S-EWC | $\alpha = 0.01, \beta_I = 0.999, \beta_w = 0.999, \kappa = 1.0$ |
| | S-MAS | $\alpha = 0.01, \beta_I = 0.999, \beta_w = 0.999, \kappa = 10.0$ |
| | S-SI | $\alpha = 0.01, \beta_I = 0.9, \beta_w = 0.9, \kappa = 0.1$ |
| | S-RWalk | $\alpha = 0.01, \beta_I = 0.9, \beta_w = 0.999, \kappa = 0.1$ |
| | UPGD-W | $\alpha = 0.01, \sigma = 0.001, \beta_u = 0.9, \lambda = 0.0$ |
| | N-UPGD-W | $\alpha = 0.01, \sigma = 0.01, \beta_u = 0.999, \lambda = 0.001$ |
| **Label-permuted *mini*-ImageNet** | SGDW | $\alpha = 0.01, \lambda = 0.001$ |
| | S&P | $\alpha = 0.01, \sigma = 0.01, \lambda = 0.001$ |
| | PGD | $\alpha = 0.01, \sigma = 0.01$ |
| | AdamW | $\alpha = 0.0001, \beta_1 = 0.9, \beta_2 = 0.9999, \epsilon = 10^{-8}, \lambda = 0.1$ |
| | S-EWC | $\alpha = 0.01, \beta_I = 0.999, \beta_w = 0.999, \kappa = 1.0$ |
| | S-MAS | $\alpha = 0.01, \beta_I = 0.999, \beta_w = 0.999, \kappa = 10.0$ |
| | S-SI | $\alpha = 0.001, \beta_I = 0.9, \beta_w = 0.99, \kappa = 0.01$ |
| | S-RWalk | $\alpha = 0.01, \beta_I = 0.99, \beta_w = 0.999, \kappa = 0.1$ |
| | UPGD-W | $\alpha = 0.01, \sigma = 0.001, \beta_u = 0.9, \lambda = 0.0$ |
| | N-UPGD-W | $\alpha = 0.01, \sigma = 0.01, \beta_u = 0.999, \lambda = 0.0001$ |

## J   THE QUALITY OF THE APPROXIMATED UTILITY

Fig. 12 shows the Spearman correlation at every time step with different activation functions. An SGD learner with a step size of $0.01$ used a single hidden layer network containing 50 units with ReLU activations (Nair & Hinton 2010) and Kaiming initialization (He et al. 2015). The network has five inputs and a single output. The target of an input vector is the sum of two inputs out of the five inputs, where the inputs are sampled from $U[-0.5, 0.5]$.

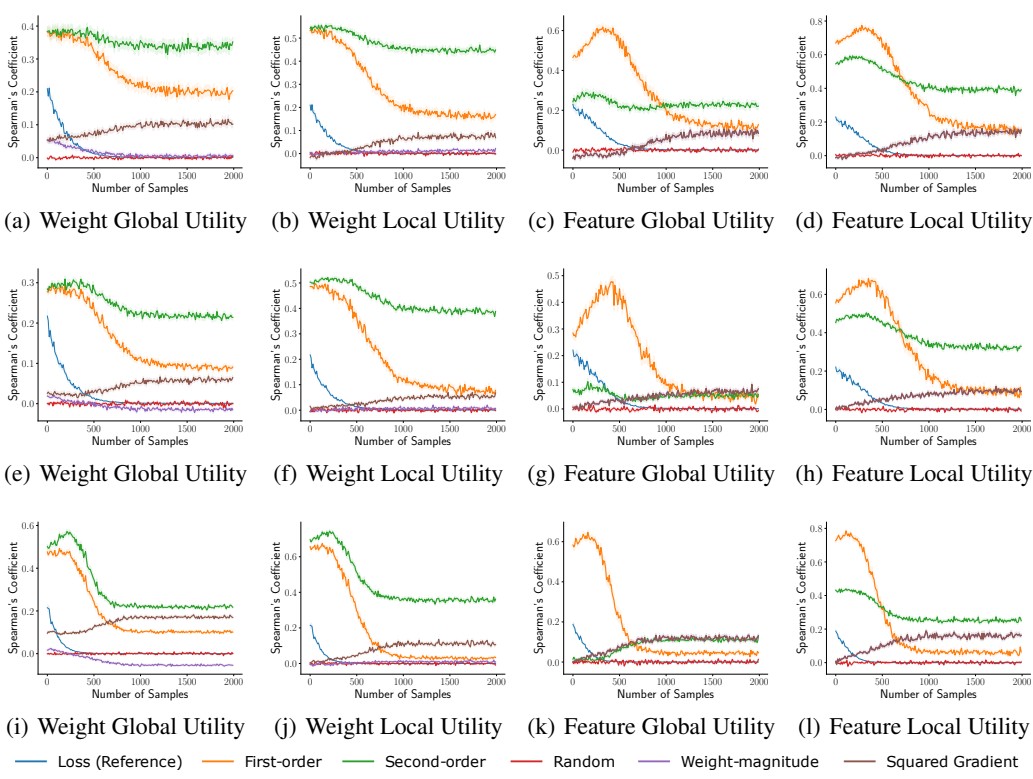

(a) Weight Global Utility   (b) Weight Local Utility   (c) Feature Global Utility   (d) Feature Local Utility

(e) Weight Global Utility   (f) Weight Local Utility   (g) Feature Global Utility   (h) Feature Local Utility

(i) Weight Global Utility   (j) Weight Local Utility   (k) Feature Global Utility   (l) Feature Local Utility

— Loss (Reference)   — First-order   — Second-order   — Random   — Weight-magnitude   — Squared Gradient

Figure 12: Spearman correlation between the true utility and approximated utilities. The activations used in the first, second, and third rows are ReLU, LeakyReLU, and Tanh. The shaded area represents the standard error.

## K   ADDITIONAL EXPERIMENTS

### K.1   MORE DIAGNOSTIC STATISTICS CHARACTERIZING SOLUTION METHODS

Here, we provide more diagnostic statistics for our methods Fig. 13, Fig. 14, Fig. 15, and Fig. 16.

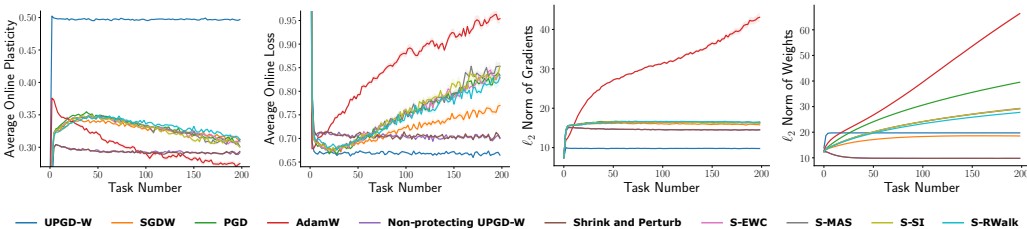

— UPGD-W   — SGDW   — PGD   — AdamW   — Non-protecting UPGD-W   — Shrink and Perturb   — S-EWC   — S-MAS   — S-SI   — S-RWalk

Figure 13: Additional diagnostic statistics of methods on Input-permuted MNIST. The average plasticity, the average online loss, the $\ell_1$-norm of gradients, and the $\ell_2$-norm of weights are shown.

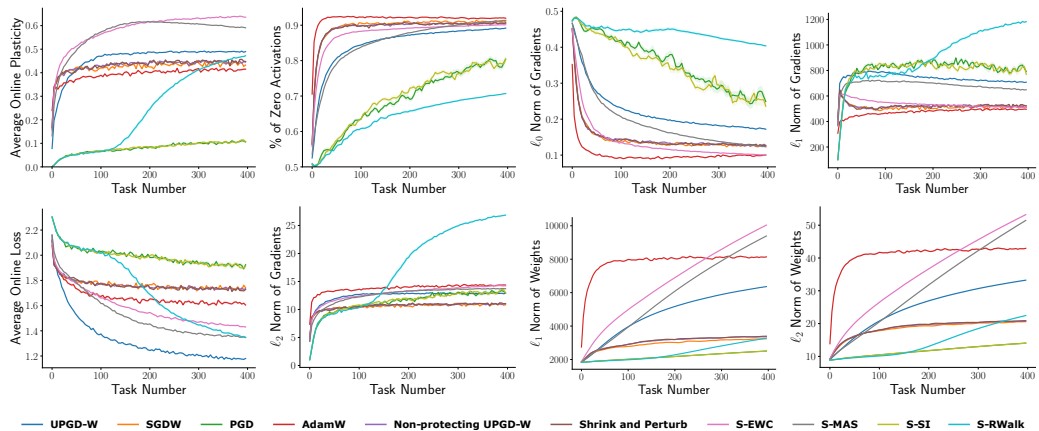

Figure 14: Diagnostic statistics of methods on Label-permuted CIFAR10. The average plasticity, percentage of zero activations, $\ell_0$, $\ell_1$ and $\ell_2$-norm of gradients, the average online loss, the $\ell_1$-norm and $\ell_2$-norm of the weights are shown. The shaded area represents the standard error.

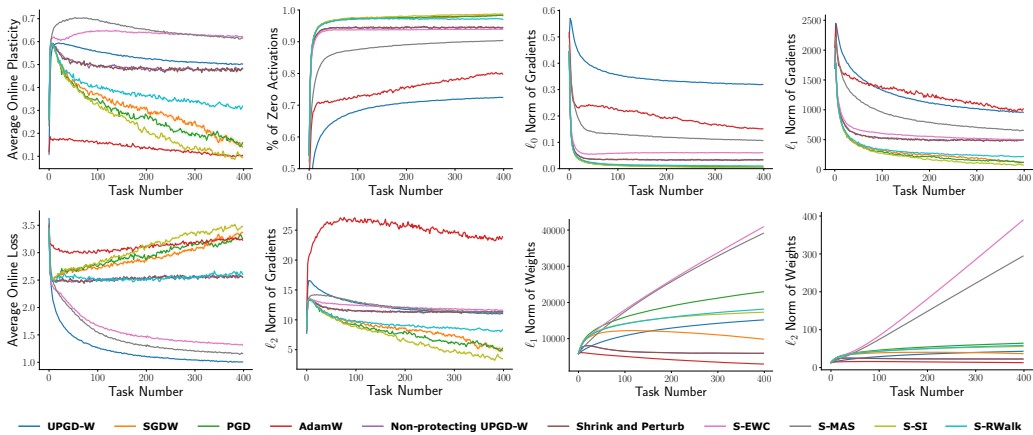

Figure 15: Diagnostic statistics of methods on Label-permuted EMNIST. The average plasticity, percentage of zero activations, $\ell_0$, $\ell_1$ and $\ell_2$-norm of gradients, the average online loss, the $\ell_1$-norm and $\ell_2$-norm of the weights are shown. The shaded area represents the standard error.

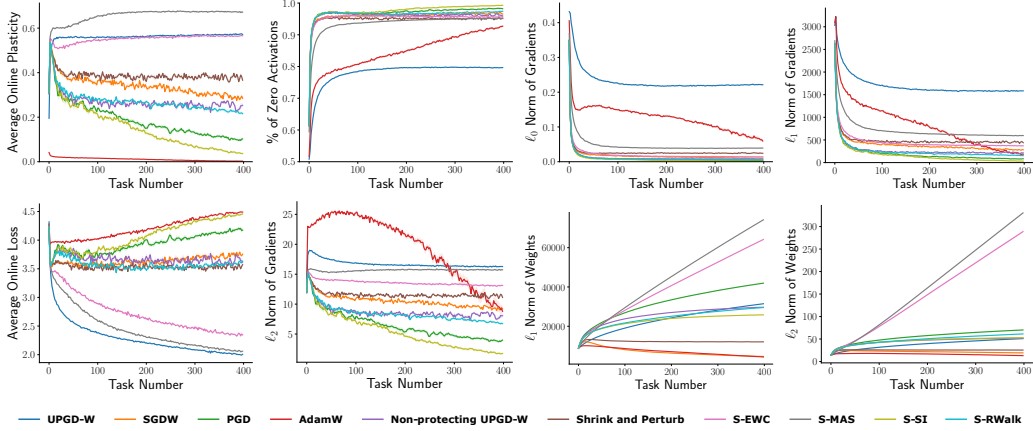

Figure 16: Diagnostic Statistics of methods on Label-permuted *mini*-ImageNet. The average plasticity, percentage of zero activations, $\ell_0$, $\ell_1$ and $\ell_2$-norm of gradients, the average online loss, the $\ell_1$-norm and $\ell_2$-norm of the weights are shown. The shaded area represents the standard error.

## K.2 COMPUTATIONAL TIME FOR MAKING UPDATES

Here, we conduct a small experiment to determine the computation overhead by each method. We are interested in the computational time each algorithm needs to perform a single update. Fig. 17 shows the computational time needed to make a single update using a batch of 100 MNIST samples. The results are averaged over 10000 updates. Each learner used a multi-layered network of $1024 \times 512$ units. The error bars represent one standard error.

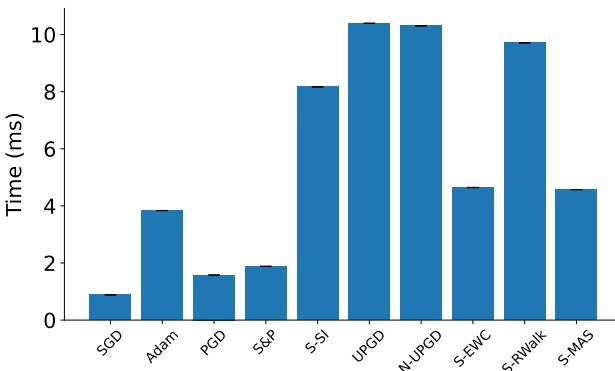

Figure 17: Computational time in milliseconds needed to make a single update. N-UPGD is short for Non-protecting UPGD.

## K.3 FEATURE-WISE AND WEIGHT-WISE UPGD ON THE INPUT-PERMUTED MNIST

We repeat the experiment on Input-permuted MNIST but with feature-wise approximated utility. In addition, we show the results using the local approximated utility for both weight-wise and feature-wise UPGD in Fig. 18. The weights are initialized by Kaiming initialization (He et al. 2015). A hyperparameter search was conducted (see Table 1), and the best-performing hyperparameter set was used for plotting.

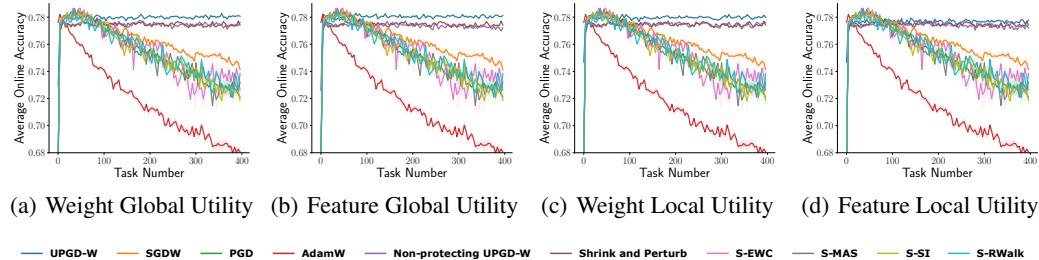

(a) Weight Global Utility   (b) Feature Global Utility   (c) Weight Local Utility   (d) Feature Local Utility

Figure 18: Performance of methods on Input-Permuted MNIST. First-order utility traces are used for UPGD and Non-protecting UPGD. Results are averaged over 10 runs.

## K.4 FEATURE-WISE AND WEIGHT-WISE UPGD ON LABEL-PERMUTED EMNIST

We repeat the experiment on Label-permuted MNIST but with feature-wise approximated utility. In addition, we show the results using the local approximated utility for both weight-wise and feature-wise UPGD in Fig. 19. The weights are initialized by Kaiming initialization (He et al. 2015). A hyperparameter search was conducted (see Table 1), and the best-performing hyperparameter set was used for plotting.

## K.5 LOCAL WEIGHT-WISE UPGD ON LABEL-PERMUTED CIFAR-10

We repeat the experiment on Label-permuted CIFAR-10 but with the local approximated utility. Fig. 20 shows the results for UPGD using local and global approximated utilities. The weights are

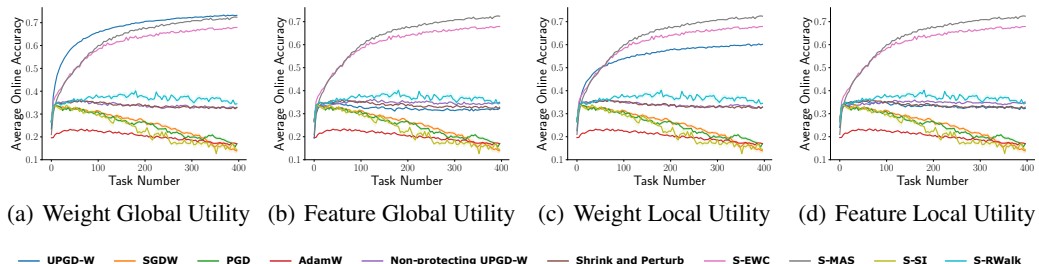

Figure 19: Performance of methods on Label-permuted EMNIST. First-order utility traces are used for UPGD and Non-protecting UPGD. Results are averaged over 10 runs.

initialized by Kaiming initialization (He et al. 2015). A hyperparameter search was conducted (see Table 1), and the best-performing hyperparameter set was used for plotting.

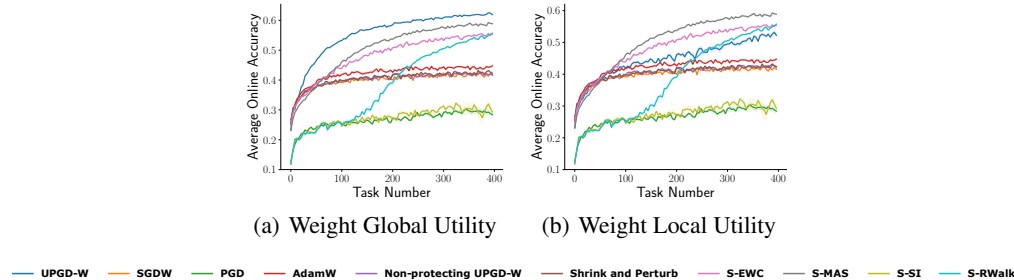

Figure 20: Performance of methods on Label-permuted CIFAR-10. First-order utility traces are used for UPGD and Non-protecting UPGD. Results are averaged over 10 runs.

## L  ABLATION ON THE COMPONENTS OF UPGD-W

We conducted an ablation study on the components of UPGD-W: weight decay, weight perturbation, and utility gating. Starting from SGD, we add each component step by step until we reach UPGD-W. Fig. 21 shows the performance of learners on Input-permuted MNIST, Label-permuted CIFAR10, Label-permuted EMNIST, and Label-permuted mini-ImageNet. We notice that both weight perturbation and weight decay separately improve SGD performance. Still, the role of weight decay seems to be more important in Input-permuted MNIST and Label-permuted mini-ImageNet. Notably, the combination of weight decay and weight perturbation makes the learner maintain its performance. When utility gating is added on top of weight decay and weight perturbation, the learner can improve its performance continually in all label-permuted problems and slightly improve its performance on input-permuted MNIST.

We also conducted an additional ablation in Fig. 22 where we start from UPGD-W and remove each component individually. This ablation bolsters the contribution of utility gating more. Using utility gating on top of SGD allows SGD to maintain its performance instead of dropping on input-permuted MNIST and improves its performance continually on label-permuted problems. The role of weight decay and weight perturbation is not significant in label-permuted problems, but including both with utility gating improves performance and plasticity on input-permuted MNIST.

## M  RELATIONSHIP TO GENERATE-AND-TEST METHODS

The generate-and-test method (Mahmood & Sutton 2013) is a method that finds better features using search, which, when combined with gradient descent (see Dohare et al. 2023a), is similar to a feature-wise variation of our method. However, this method only works with networks with single-hidden layers in single-output regression problems. It uses the weight magnitude to determine feature utilities; however, it has been shown that weight magnitude is not suitable for other problems,

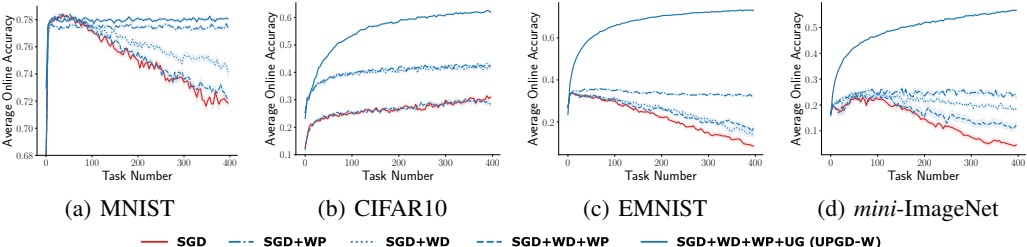

Figure 21: Ablation on the components of UPGD-W: Weight Decay (WD), Weight Perturbation (WP), and Utility Gating (UG) shown on Input-permuted MNIST, Label-permuted EMNIST, Label-permuted CIFAR10, and Label-permuted *mini*-ImageNet. A global first-order utility trace is used. Results are averaged over 10 runs. The shaded area represents the standard error.

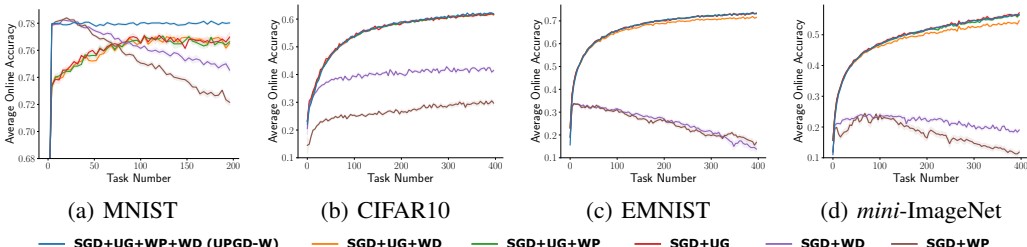

Figure 22: Ablation on the components of UPGD-W: Weight Decay (WD), Weight Perturbation (WP), and Utility Gating (UG) shown on Input-permuted MNIST, Label-permuted EMNIST, Label-permuted CIFAR10, and Label-permuted *mini*-ImageNet, starting from UPGD-W and removing each component individually. SGD+WD and SGD+WP are added as baselines that do not use utility gating. A global first-order utility trace is used. Results are averaged over 10 runs. The shaded area represents the standard error.

such as classification (Elsayed 2022). On the contrary, our variation uses a better notion of utility that enables better search in the feature space and works with arbitrary network structures or objective functions so that it can be seen as a generalization of the generate-and-test method.

## N    FUTURE WORKS

Arguably, one limitation of our approach is that it measures the weight utility, assuming other weights remain unchanged. A better utility would include interaction between weight variations, which is left for future work. One desirable property of continual learners is the ability to modify their hyperparameter, allowing for greater adaptation to changes. Although our method does not require intensive hyperparameter tuning, it still requires some level of tuning, similar to other methods, which may hinder its ability in true lifelong learning. A promising direction is to use a generate-and-test approach that continually learns the best set of hyperparameters. In our evaluation, we assumed that the non-stationary target function is piece-wise stationary. Studying the effectiveness of UPGD against hard-to-distinguish non-stationarities would be an interesting future direction.

