# OpenReview forum: "Addressing Loss of Plasticity and Catastrophic Forgetting in Continual Learning"
_ICLR.cc/2024/Conference — ICLR 2024 poster_

### Official Review · Reviewer_KmBd · 2023-10-23

**Soundness:** 3 good
**Presentation:** 3 good
**Contribution:** 3 good
**Rating:** 6
**Confidence:** 4

**Summary:**

This paper introduces a novel approach called Utility-based Perturbed Gradient Descent (UPGD) to address catastrophic forgetting and loss of plasticity in neural networks. UPGD combines gradient updates with a mask to protect useful weights from being forgotten and reuse less useful weights. The paper also proposes metrics to evaluate loss of plasticity and catastrophic forgetting. Empirically, the method outperforms existing methods in streaming learning problems in terms of retaining plasticity and avoiding catastrophic forgetting.

**Strengths:**

**Originality**
Conceptually, the problem addressed by the authors of avoiding forgetting while retaining plasticity in streaming learning settings remains underexplored. The specific method proposed by the authors is relatively straightforward and conceptually similar to prior approaches; however, it empirically outperforms prior methods as the experimental results demonstrate.

**Quality**
The convergence guarantee results are valuable. The experiments are generally comprehensive and well-conducted. Assessing the quality of the approximated utilities in section 4.1 is
of critical importance, and the results are convincing. Conducting miniImagenet scale experiments is a solid addition to the experimental section. The ablation study in Figure 8 is also insightful.

**Clarity**
The writing is generally clear and the figures are well-illustrated.

**Significance**
Overall, the paper addresses a major issue in the field of streaming learning. Given that the paper doesn't investigate the theoretical properties of UPGD, the significance of the paper hinges on the strength of the empirical results.

**Weaknesses:**

Since the proposed method lacks theoretical performance guarantees, its empirical performance is critical. The authors have generally done a good job demonstrating that UPGD avoids forgetting and maintains plasticity; however, a few concerns remain:

- It appears that S-EWC does not have too much of a gap with UPGD judging from figure 7: it entirely avoids catastrophic forgetting, and the only setting where it loses plasticity where UPGD does not is on MNIST
- S-MAS outperforms UPGD on miniImagenet at the end of training, and does not have a large gap overall
- The ablation of figure 8 checks the contribution of each component of UPGD sequentially as they are added to regular SGD. Ideally, the ablation would study how each component affects UPGD when they are *individually* removed (e.g. UPGD without WP).

**Minor comments**
Figure 7 is referred to before Figure 6; ideally, their order would be swapped.
I see in Section 4 that the results are averaged over 20 trials, but the meaning of the error margins in some of the figures is not made clear (e.g. figure 2). I would also suggest increasing the number of trials to smooth out the curves if possible.

**Questions:**

Is it possible to show theoretical performance guarantees for UPGD? For instance, can the approximation error of equation 2 be bounded? Alternatively, if the true utilities are used in equation 3, is it possible to derive some guarantees against forgetting or loss of plasticity?

How much more significantly does UPGD improve upon baselines S-EWC and S-MAS?

How does UPGD-W perform with WP and WD removed individually?

---

> ### Author Response · Authors · 2023-11-14
>
> We would like to extend our sincere gratitude to the reviewer for the time and effort invested in reviewing our paper. We are also grateful to read that the reviewer thinks our work is significant and novel in an underexplored area with comprehensive and well-conducted experiments and clearly written paper. Here, we give our response to the points the reviewer mentioned.
>
> &nbsp;
>
> > It appears that S-EWC does not have too much of a gap with UPGD judging from Figure 7: it entirely avoids catastrophic forgetting, and the only setting where it loses plasticity where UPGD does not is on MNIST
>
> While S-EWC is effective in catastrophic forgetting evaluation, it suffers from loss of plasticity, which was the aspect we evaluated in the input-permuted MNIST task. Only UPGD addressed both loss of plasticity and catastrophic forgetting, which is shown by its performance on both evaluations.
>
> &nbsp;
>
> > S-MAS outperforms UPGD on miniImagenet at the end of training, and does not have a large gap overall
>
> While we do not expect a new method to outperform in every single task, UPGD still achieves higher average accuracy in the first half of the experiment. It is desirable in streaming learning where the online accuracy at each time step is more important than the accuracy only at the end of training.
>
> &nbsp;
>
> > The ablation of figure 8 checks the contribution of each component of UPGD sequentially as they are added to regular SGD. Ideally, the ablation would study how each component affects UPGD when they are individually removed (e.g. UPGD without WP).
>
> We conducted the requested ablation. Please see this figure [[link]](https://drive.google.com/file/d/15aeYYq3QjeEgAbeFVeqzSNT5Ms2QvSYr/view?usp=sharing). The requested ablation bolsters the contribution of utility gating even more. Using utility gating on top of sgd makes sgd maintain its performance instead of dropping on input-permuted MNIST and makes sgd improve its performance continually on label-permuted problems. The role of weights decay and weight perturbation is not significant in label-permuted problems but including both with utilty gating improves performance and plasticity on input-permuted MNIST.
>
> &nbsp;
>
> > Minor comments Figure 7 is referred to before Figure 6; ideally, their order would be swapped. I see in Section 4 that the results are averaged over 20 trials, but the meaning of the error margins in some of the figures is not made clear (e.g. figure 2). I would also suggest increasing the number of trials to smooth out the curves if possible.
>
> Thank you for pointing out the order of mention for Figures 6 and 7; we will make the necessary amendments. The error margins represent standard error. Our results are averaged over 20 trials. We use non-overlapping intervals to average over. Therefore, our plots can have variabilities even though we have a large number of trials. We don’t average by using an exponential moving average, which will smoothen out the curves since we are interested in what each point in the plot means. For example, each point in Figure 3 represents the average performance over a single task. Smoothing over time wouldn’t give the same meaning.
>
> &nbsp;
>
> **Answers the Questions:**
>
> > Is it possible to show theoretical performance guarantees for UPGD?
>
> We think this would be interesting direction that is left for future work.
>
> > How much more significantly does UPGD improve upon baselines S-EWC and S-MAS?
>
> Based on our results, we can see that UPGD outperforms S-EWC in all experiments and outperforms S-MAS in 3 out of 4 experiments.
>
> > How does UPGD-W perform with WP and WD removed individually?
>
> Please see this figure [[link]](https://drive.google.com/file/d/15aeYYq3QjeEgAbeFVeqzSNT5Ms2QvSYr/view?usp=sharing), which includes the requested ablation.

---

> > ### Author Response · Authors · 2023-11-23
> >
> > Since the discussion period is ending soon, we are following up to see if our response addressed the reviewer’s concerns. We would greatly appreciate the reviewer’s acknowledgment.

---

### Official Review · Reviewer_3zCE · 2023-10-30

**Soundness:** 4 excellent
**Presentation:** 3 good
**Contribution:** 1 poor
**Rating:** 3
**Confidence:** 4

**Summary:**

This work proposes to modify stochastic gradient descent (SGD) to overcome forgetting and promote plasticity in continual learning. These goals are achieved by masking out the parameters with high utility and perturbing gradient direction by Gaussian noise. For utility computation, the authors propose an approximate but efficient scheme based on second-order Taylor expansion of the loss. Experiments demonstrate that (i) the proposed utility approximation is more accurate than simple baselines such as weight magnitude, (ii) it maintains plasticity, (iii) plasticity and accuracy are in general correlated, (iv) the method tends to forget less than baselines, and (v) it simultaneously promote plasticity and prevents forgetting.

**Strengths:**

This paper is written well. The notation is okay and the mathematical derivations seem correct. The baseline methods are clearly outperformed and the experiments verify the central claim of the paper.

**Weaknesses:**

Although continual learning is an important machine learning challenge, I feel the paper suffers from significant weaknesses:

  - First and foremost, I do not think this paper makes a significant contribution. The methodology is incremental in that it combines two well-known ideas (perturbed gradient descent + keeping active neurons unchanged).
  - Second, it is tested on very toy setups. The experiments are not convincing enough to show the applicability of the method to interesting real-world setups. For instance, I am not sure the networks achieve similar plasticity if tested on, e.g., webcam data instead of MNIST, where the feature space is a lot richer and hence plasticity is much more difficult.
  - Third, theoretical properties/implications of the method should be carefully examined.
    - For instance, the Taylor expansion would only hold if $W_{l,i,j}$ are infinitesimally small. We do not know in general if this holds or not. I suggest the paper should include a (preferably rigorous) discussion on this.
    - Likewise, gradient descent is no longer steepest descent but some approximation to it. Investigating why it works is important. As shown by the results, no collapse occurs but again, I wonder how this translates into more challenging settings where utilities of most parameters are high.

**Questions:**

Here I list my questions as well as suggestions:

- It would be better if Label-Permuted EMNIST was described before the results are discussed in paragraph 5.
- _Although a few methods address both issues simultaneously, such methods expect known task boundaries, maintain a replay buffer, or require pretraining, which does not fit streaming learning._ <--- reference needed for this claim.
- What does "a Hessian diagonal approximation in linear complexity" mean? Linear in the number of parameters?
- It would be better if the main text included details on the "utility propagation theorem".
- It would be better if the descriptions of the tasks/datasets (e.g. Input-Permuted MNIST in section 4.2) were given before the details.
- Does "each learner is trained for 1M samples, one sample each time step" mean gradient descent using one sample only? Is this realistic?

---

> ### Author Response · Authors · 2023-11-14
>
> We would like to extend our sincere gratitude to the reviewer for the time and effort invested in reviewing our paper. We are also grateful to read that the reviewer thinks our paper is well-written and that our empirical evaluation verifies the central claim of the paper. Here, we give our response to the points the reviewer mentioned.
>
> &nbsp;
>
> > I do not think this paper makes a significant contribution. The methodology is incremental in that it combines two well-known ideas (perturbed gradient descent + keeping active neurons unchanged).
>
> We respectfully disagree with the reviewer. First, our method does not work by keeping active neurons unchanged. Our method uses a utility-gating mechanism to protect useful weights from change and perturb less useful ones. This idea is compatible with weight perturbation and weight decay, which makes the method stronger. Second, our novel utility-gating mechanism presents an instance of a new class of methods that has not been introduced before. We kindly ask the reviewer to refer us to works to which UPGD is considered incremental.
>
> &nbsp;
>
> > It is tested on very toy setups.
>
> Our tasks are widely used in continual learning for evaluating forgetting and plasticity (see [1,2,3,4]), where input-permuted tasks were used for evaluating plasticity and output-permuted tasks were used for evaluating forgetting. We certainly acknowledge the reviewer’s valuable suggestion for real-world setups, such as webcam data. However, such benchmarks are unavailable, especially for streaming learning, due to its complexity, and requires a separate dedicated paper. As you recognize that our experiments verify the paper’s central claim and reviewer KmBd highlights their scale, solidity, and comprehensiveness, we believe they sufficiently demonstrate the core contribution of our work.
>
> &nbsp;
>
> > Third, theoretical properties/implications of the method should be carefully examined.
>
> We provided theoretical work in the Appendix. Please refer to Appendix A for convergence analysis, Appendix B for how utilities can be propagated in the network, and Appendix D for the connection between weight and feature utilities.
>
> Regarding the comment on Taylor expansion, our first and second-order Taylor approximations require ignoring higher-order terms, which is common and valid for our case when weights are proper fractions, making them sufficiently small. Our empirical evaluation shows the effectiveness of this approximation in continual learning. For future work, one can consider a different utility definition based on the change in loss from sample to sample, which depends on the difference between the values of the weight from two consecutive time steps, making the approximation more accurate (see [5] and Appendix M).
>
> &nbsp;
>
> > gradient descent is no longer steepest descent but some approximation to it. Investigating why it works is important.
>
> In Appendix A, we provided convergence analysis for our method, where we showed that our method converges to a stationary point, similar to other methods (e.g., SGD). Our empirical evaluation in Figure 9 shows that UPGD also outperforms SGD in stationary problems.
>
> &nbsp;
>
> **Answers to the Questions:**
>
> First, we thank the reviewer for the suggestions. We will incorporate them in our revision.
>
> > reference needed for this claim.
>
> Thank you for letting us know. We will add these references in the paper revision (see [6,7]).
>
> > What does "a Hessian diagonal approximation in linear complexity" mean? Linear in the number of parameters?
>
> This is correct: linear in the number of parameters.
>
> > Does "each learner is trained for 1M samples, one sample each time step" mean gradient descent using one sample only? Is this realistic?
>
> That is correct. The update is made based on one sample at a time. This is often used as the more challenging constraint of streaming learning [e.g., 8]
>
> &nbsp;
>
> *References:*
> \
> [1]. Lyle, C., Zheng, Z., Nikishin, E., Pires, B. A., Pascanu, R., & Dabney, W. (2023). Understanding plasticity in neural networks.
> \
> [2]. Dohare, S., Hernandez-Garcia, J. F., Rahman, P., Sutton, R. S., & Mahmood, A. R. (2023). Maintaining Plasticity in Deep Continual Learning.
> \
> [3]. He, X., Sygnowski, J., Galashov, A., Rusu, A. A., Teh, Y. W., & Pascanu, R. (2019). Task agnostic continual learning via meta learning.
> \
> [4]. Kumar, S., Marklund, H., & Van Roy, B. (2023). Maintaining plasticity via regenerative regularization.
> \
> [5]. Zenke, F., Poole, B., & Ganguli, S. (2017, July). Continual learning through synaptic intelligence.
> \
> [6]. Hayes, T. L., Kanan, C. (2022). Online continual learning for embedded devices.
> \
> [7]. Van de Ven, G. M., Siegelmann, H. T., & Tolias, A. S. (2020). Brain-inspired replay for continual learning with artificial neural networks.
> \
> [8] Sahoo, D., Pham, Q., Lu, J., & Hoi, S. C. (2017). Online deep learning: Learning deep neural networks on the fly.

---

> > ### Comment · Reviewer_3zCE · 2023-11-22
> > **my response to author response**
> >
> > Thanks for the author response. My main concern is novelty, which I feel is still not addressed in the author response. As such, the methodology has two pillars: perturbed masked gradient descent and masking based on utility. Concerning the first pillar, please see "Masked Training of Neural Networks with Partial Gradients" paper and all the references therein. Further, see "Learning where to learn: Gradient sparsity in meta and continual learning" paper for yet another presentation of gradient masking for increasing plasticity. About the second pillar, my main concern is that the Taylor approximation does not hold unless $W_{l,i,j}$ is infinitesimally small. I really think this is a strong assumption that does not necessarily generalize, although I acknowledge that the method works fine in practice.

---

> > > ### Author Response · Authors · 2023-11-23
> > >
> > > We thank the reviewer for reading our response and engaging in the discussion.
> > >
> > > > Taylor approximation does not hold unless $W_{l,i,j}$ is infinitesimally small.
> > >
> > > It is true that the Taylor approximation produces small errors when some conditions are satisfied. In our case, weights only need to be a small fraction, not infinitesimally small, to drop the higher-order terms, which we also mentioned above. Typically, the vast majority of the weights in a network are small fractions. For example, the last plot of Figure 5 in the paper shows SGDW has an L1 weight norm of ~6000 for a network with 282160 weights, giving an average weight magnitude of ~0.02. Moreover, we use weight decay, which also helps keep the weights small. Thanks for pointing this out. We will clarify it further in our paper.
> > >
> > > > masked gradients
> > >
> > > We acknowledge that there are similarities between the idea of masking and gating. Using gating mechanisms to regulate some quantity is a simple and old idea that has been used in different contexts in different ways (e.g., LSTM [1] and GRU [2] in the context of recurrent learning). Hence, we believe how such gating mechanisms are used is important and can still constitute novelty. And the idea of using a gating mechanism based on utility in our work is novel. The purpose of our utility gating is also different from that of masking in the reviewer’s cited works, where the former protects important weights, and the latter achieves sparsity.
> > >
> > > We also ask the reviewers if they would agree to raise the score if we addressed the concerns at least to some extent, given that our work shows significant progress in addressing two important issues of continual learning simultaneously. At the same time, we would be grateful if the reviewer could suggest what more would be needed to make this work acceptable to the reviewer and accessible to a broader audience. We thank the reviewer again for engaging in this discussion and are looking forward to further dialogue on this matter.
> > > \
> > > \
> > > References:
> > > \
> > > [1] Hochreiter, S., & Schmidhuber, J. (1997). Long short-term memory. Neural Computation, 9(8), 1735-1780.
> > > \
> > > [2] Cho, K., van Merrienboer, B., Gulcehre, C., Bougares, F., Schwenk, H., & Bengio, Y. (2014). Learning phrase representations using RNN encoder-decoder for statistical machine translation. Conference on Empirical Methods in Natural Language Processing.

---

### Official Review · Reviewer_y5kB · 2023-10-30

**Soundness:** 3 good
**Presentation:** 3 good
**Contribution:** 3 good
**Rating:** 6
**Confidence:** 3

**Summary:**

The paper proposes a measure of weight utility of weights in neural networks for given loss and using it to modify a gradient-based weight update in networks to alleviate the problem of catastrophic forgetting.  The authors identify two fundamental aspect of catastrophic forgetting - the forgetting aspect (not losing what the network already know) and plasticity aspect (ability to learn new concepts).  The proposed method is meant to address two problems at the same time, preseving high utility weights with no modifications (to prevent forgetting) while randomly perturbing low utility weights to "encourage" them to participate in the computations related to new tasks (plasticity).  Empirical evaluations show solid performance of the proposed method according to the forgetting and plasticity metrics newly defined by the authors.

**Strengths:**

The proposed rule is straight forward.

Computational complexity of the evaluation of true utility is well addressed making the method practical.

Empirical evidence provided shows the proposed rule is effective for alleviation of catastrophic forgetting.

Decomposing the catastrophic forgetting problem into two aspects: forgetting and plasticity, seems very sensible.

Proposed measures of plasticity and forgetting seem sensible.

The paper is well written.

**Weaknesses:**

Though empirical evidence provided in the paper suggest it does (in that it works), I am not sure that the proposed definition of weight utility make sense.  The power of neural networks (and the problem of the interpretation of its computation) is its distributed computation.  Utility of an individual weight is almost always nothing - in fact, quite often any particular weight, sometimes even large number of weights, can be taken out of the network, with little impact on performance.  So, it's more about combinations of weights working together...and the proposed utility doesn't measure that.  I understand that evaluating utility of combinations of weights is intractable, but I worry that this simplification, of judging utility of each weight in isolation, is encouraging less distributed representation, which might come with a penalty in performance.

Fundamentally, on the forgetting front, the proposed method is just another weight consolidation method, and it's a bit hard to believe it beats Elastic Weight Consolidation.  It am not 100% sure that the proposed method doesn't favour plasticity over forgetting nor that the forgetting evaluation isn't biased towards methods that favour plasticity (see questions below).

**Questions:**

Though I understand (and like) in principle what the utility-based update is supposed to do, I can't quite understand why it actually works.  The proposed measure of the utility of parameters is a measure with respect to the loss on the new input/output pair.  If this pair comes from a new task, how does measuring utility of the model parameters with respect to the loss of this new task have bearing on the utility of the parameters for the old tasks?  Just because utility of a given weight is, say, low for the current sample, it doesn't mean it's low for previous samples.  It seems to me that the proposed method would score high on plasticity (it finds available weights for new task)...but I don't see how it protects against forgetting, in principle, though if we are to talk about empirical evidence...  I don't understand how 4.3 measures catastrophic forgetting.  Permuting labels of CIFAR10 with the new tasks suggests to me that it's all about plasticity again.  Shouldn't it be an experiment, where labels are kept intact, but new tasks are added...and previous tasks examples are not used?  Am I missing something about how experiments reported in 4.3 are done?

Why are the accuracy results of training on CIFAR-10 and EMNIST so poor in Figure 6?  State of the art CIFAR-10 is close (or above) 90%.  Something close to 80% would be probably still acceptable...but 60% is quite poor.  I am not exactly sure what EMNIST variant entails, but is 70% accuracy a good accuracy for this dataset?  It is often easy to shown improvements of something at the low end of the models' performance, but that doesn't always translate to same effect at the high (or close to) end of the models' performance...and in the end, the latter is what we really care about.  So, does the proposed method prevent forgetting at the high end, when model is performing at or reasonably close to state of the art?

This is not a massive issue, but does the per batch normalisation of utility make the performance of the method variable with different  mini-batch size settings?

---

> ### Author Response · Authors · 2023-11-14
>
> We would like to extend our sincere gratitude to the reviewer for the time and effort invested in reviewing our paper. We are also grateful to read that the reviewer thinks our method is straightforward and practical, our metrics are reasonable, our paper is well-written, and our method has solid performance. Here, we give our response to the points the reviewer mentioned.
>
> &nbsp;
>
> > Utility of an individual weight is almost always nothing...it's more about combinations of weights working together...and the proposed utility doesn't measure that
>
> We agree with the reviewer that a better utility definition would include the interaction between weights (as we pointed out in Appendix M). However, including such interactions makes the method intractable, as the reviewer mentioned. Most methods addressing forgetting (e.g., EWC) use individual weight measures, and yet they are effective. While we are interested in learning about the kind of representations learned by each method, we believe this merits a dedicated paper to address fully. From our experiment, however, we found that UPGD improves performance over SGD even in the stationary case (see Figure 9 in Appendix F), likely due to finding better representations.
>
> &nbsp;
>
> > the proposed method is just another weight consolidation method, and it's a bit hard to believe it beats Elastic Weight Consolidation.
>
> While there are strong connections and similarities between utility-gating and regularization-based methods, our approach differs from Elastic Weight Consolidation (EWC) in two major ways: 1) In EWC, a regularization term is essential, while our method has no such term., and 2) our proposed utility measure is different than EWC's squared gradient metric. The results from Figure 2 show the empirical superiority of our utility measure over the squared gradients, which partially explains the performance difference. More future theoretical work is needed to link between these two classes of methods.
>
> &nbsp;
>
> > It am not 100% sure that the proposed method doesn't favour plasticity over forgetting nor that the forgetting evaluation isn't biased towards methods that favour plasticity.
>
> In our catastrophic forgetting (CF) evaluation, methods that only address CF can achieve high performance (e.g., S-MAS) since the problem is designed to favor methods that keep useful representations and improve on them continually. We do not think our CF evaluation is biased towards plasticity. Plasticity-inducing methods (e.g., Shrink and Perturb) achieve low performance in CF evaluation.
>
> &nbsp;
>
> **Answers:**
>
> > Just because utility of a given weight is, say, low for the current sample, it doesn't mean it's low for previous samples.
>
> The utility measure we use is based on an exponential moving average of the sample utilities, as we explain in the last sentence of the second paragraph of section 4.2. Thus, it does not suffer from the issue the reviewer mentioned about sample instantaneous utilities.
>
> > how 4.3 measures catastrophic forgetting. Permuting labels of CIFAR10 with the new tasks suggests to me that it's all about plasticity.
>
> Section 4.3 indeed evaluates catastrophic forgetting, as previously learned representations transfer and remain useful from one task to another. Hence, methods that forget previously learned representations should suffer in performance. Our results also confirm that methods that protect useful representations and improve them with more data achieve higher performance (e.g., S-MAS). A method that does not protect useful representations (e.g., SGD) forgets them by overwriting useful information and, therefore, does not improve its performance continually.
>
> > Why are the accuracy results of training so poor?
>
> In our streaming non-stationary problems, achieving state-of-the-art results as in stationary settings is unexpected due to their incomparability. Please see Figure 2b (rightmost of the three) in [1], where accuracy decreases with increased permutation frequency: from 99% (permuting every 1M samples) to about 88% (every 10K). Our experiment, with permutations every 5K samples, achieves a reasonable best accuracy of around 78%, aligning with this trend.
>
> > does the proposed method prevent forgetting at the high end?
>
> In streaming learning, the evaluation is done on-the-fly at every time step, so maximizing the online performance (accuracy) at every step by reducing forgetting is desired. It’s a different problem setting from the one the reviewer described, where the final performance is what is important.
>
> > does the per batch normalisation of utility make the performance of the method variable with different mini-batch size settings?
>
> In our paper, we used a batch size of 1 (streaming case). Our method can also work with mini-batches where batch normalization might help.
>
> &nbsp;
>
> *References:*
>
> [1]. Dohare, S., Hernandez-Garcia, J. F., Rahman, P., Sutton, R. S., & Mahmood, A. R. (2023). Loss of Plasticity in Deep Continual Learning.

---

> > ### Comment · Reviewer_y5kB · 2023-11-22
> > **Thank you for response**
> >
> > I would like to thank the authors for answering my questions.  They cleared up a number of my misunderstandings, which reassures my rating of marginally above acceptance threshold.

---

> > > ### Author Response · Authors · 2023-11-23
> > >
> > > We would like to thank the reviewer for the reply. Given that our response cleared their concerns, we ask the reviewer what more, in their opinion, would be needed to make this work stronger and accessible to a broader audience.

---

### Official Review · Reviewer_XNx6 · 2023-11-05

**Soundness:** 2 fair
**Presentation:** 2 fair
**Contribution:** 3 good
**Rating:** 6
**Confidence:** 4

**Summary:**

This paper proposes Utility-based Perturbed Gradient Descent(UPGD). A modification to the vanilla gradient descent update rule that helps the model to operate in a more challenging scenario of streaming learning. The authors introduced their utility function as an importance weight for each parameter of a neural network. The authors show the effectiveness of their contribution compared to common importance assignment methods in the continual learning literature.

**Strengths:**

**Clear Structure and Writing:** The paper benefits from a clear structure and concise writing style.

**Addressing a Complex Issue:** The authors tackle an underexplored yet challenging problem, and I appreciate their efforts to address online continual learning.

**Mathematical Foundation:** The definition of utility introduced in the paper is based on simple and sound mathematical derivations.

**New metric:** The introduction of a new plasticity metric is a nice contribution to the relatively uncharted territory of streaming learning.

**Weaknesses:**

**Unscaled perturbations:** My main point of issue is the reasoning behind the perturbations in the update rule. The authors claim that by adding the perturbation we are making the unimportant weights more plastic however I am not really convinced by this explanation I believe it requires elaboration both in the rebuttal and in the paper.

Another related issue with the proposed perturbation is the fact that all of them are getting drawn from the same standard normal distribution. This design choice is strange to me since the parameters of a neural network usually differ in magnitude from layer to layer. By adding an unscaled random perturbation to all of the weights we are ignoring this scale difference which I believe is sub-optimal. I know that in the unprotected version, they are getting weighted by different values but this particular scaling is more correlating with changes in the loss value rather than the parameter magnitudes.

Highly relevant to the above issue, I believe it is also necessary to have an additional ablation study, investigating the role of having and not having the perturbations in the update rule. I also want to disentangle the effect of weight decay. The only time that UG is added in the ablation is in the presence of WD. More specifically I am curious about the following scenarios in Figure 8:

* Added ablations:
    + SGD + UG + WP + WD (present in the paper)
    + SGD + UG + WP
    + SGD + UG + WD
    + SGD + UG


**Including more diverse experiments:** Moreover, in the experiments section I believe the authors need to include more diverse experiments. All of the streaming tasks are permutations of the same task. Whether in the label or in the input space. It is not as obvious as the authors' claim that after the permutation of the input space the previously learned representations are not relevant anymore (end of page 6). In the input-permuted scenario, only the first layer needs to have significant change. This is especially true for the label-permuted tasks as the network does a good job of clustering the data up to the final FC layer. I encourage the authors to use the Cifar100 superclass dataset (or any similar sequence of tasks that does not simply rely on the permutation).

**Visualization:** Finally, I believe the visualization needs several improvements: the legends on the plot are very hard to read (Fig 2, 3, 4, 5). Some colors are similar to each other and the width of the lines in the legends is too thin. (Especially in figure 4). In Figure 7, some numbers in dark blue cells are almost impossible to read.

**Questions:**

**Q1:** Have the authors tried to use an scaled version of the perturbation that takes the magnitude of the parameters into account? (Other than the unprotected version). Also I would appreciate the if you could elaborate on the effect of perturbations.

**Q2:** Could you also explain about the average online accuracy? it is stated that "The average online accuracy is the percentage of correct predictions within each task." I cannot see the average part here. Is it calculating the accuracy on each task separately then averaging over the number of tasks?

---

> ### Author Response · Authors · 2023-11-14
>
> We would like to extend our sincere gratitude to the reviewer for the time and effort invested in reviewing our paper. We are also grateful to read that the reviewer thinks our paper tackles an underexplored but challenging problem and our paper is well-written with sound mathematical derivations. We are also happy that the reviewer appreciated our new plasticity metric contribution. Here, we give our response to the points the reviewer mentioned.
>
> &nbsp;
>
> > The authors claim that by adding the perturbation we are making the unimportant weights more plastic however, I am not really convinced
>
> Noise injection has been shown in the literature of plasticity as a way of introducing plasticity (see [1]). In Figure 3 of our work, S&P maintains its plasticity compared to SGD with weight decay, mainly because of noise perturbation (which is unscaled). While noise perturbation alone might not be effective (see PGD in the same figure), its role is pronounced when used with weight decay (S&P) or when used on a per-feature level [1]. We will add a sentence to make this clearer in our discussion of the method.
>
> &nbsp;
>
> > all of proposed perturbation are getting drawn from the same standard normal distribution...adding an unscaled random perturbation to all of the weights we are ignoring this scale difference
>
> First, we note that the standard deviation of the noise perturbation is a hyperparameter that can be tuned, which is effectively a scaling of the perturbation (see Table 2 and Table 3 for the values used). Second, we agree that noise perturbation can also use explicit layer-wise scaling. However, our utility scaling takes into account the importance of weight, meaning that weights in the earlier layers generally have higher utility and, therefore, perturbed less. Finally, we acknowledge that there are always ways to improve any method (we discussed some of them in Appendix M). Yet, our simple design was found to be very effective compared to other methods.
>
> &nbsp;
>
> > It is also necessary to have an additional ablation study, investigating the role of having and not having the perturbations in the update rule.
>
> We conducted the requested ablation. Please see this figure [[link]](https://drive.google.com/file/d/15aeYYq3QjeEgAbeFVeqzSNT5Ms2QvSYr/view?usp=sharing). The requested ablation bolsters the contribution of utility gating even more. Using utility gating on top of sgd makes sgd maintain its performance instead of dropping on input-permuted MNIST and makes sgd improve its performance continually on label-permuted problems.
>
> &nbsp;
>
> > It is not as obvious as the authors' claim that after the permutation of the input space the previously learned representations are not relevant anymore
>
> We agree with the reviewer on the comment about the phrase “previously learned representations are not relevant anymore”. We meant to use the word "features" instead of "representations". We will correct it in our revision.
>
> &nbsp;
>
> >Including more diverse experiments
>
> The problems we use are extensively adopted by the continual learning community to evaluate forgetting or plasticity in a controlled manner. We tried split-based tasks, as the reviewer kindly suggested, with CIFAR 100, but we found that these problems become very easy to the extent that all methods achieve an online accuracy of above 98% because, in streaming learning with long sequences, tasks have to re-occur which makes the problem easy (see [2]). Therefore, we chose label-permuted tasks since task re-occurrence probability becomes very small. Finally, due to the complexity and significance of designing realistic benchmarks suitable for streaming learning, we believe it merits a dedicated paper to address fully.
>
> &nbsp;
>
> > I believe the visualization needs several improvements.
>
> We thank the reviewer of this note. We will improve visualization in our revision.
>
> &nbsp;
>
> **Answers:**
>
> *A1:* The weight perturbations in UPGD are still scaled by utility. They are just scaled by the same utility the gradients are scaled by. Future work can study or improve the relative scaling between weight perturbation and gradient information.
>
> *A2:* In streaming learning, the learner receives a data stream and outputs a stream of predictions that are either correct (sample accuracy of 1) or incorrect (sample accuracy of 0). The average accuracy on a certain task is the average of these sample accuracies, which is also the percentage of correct predictions since the sample accuracies are either zero or one. We show the online average accuracy on each of the sequential tasks we have, so it is an average over each task separately.
> \
> \
> *References:*
> \
> [1]. Dohare, S., Hernandez-Garcia, J. F., Rahman, P., Sutton, R. S., & Mahmood, A. R. (2023). Loss of Plasticity in Deep Continual Learning.
> \
> [2] Lesort, T., Ostapenko, O., Rodriguez, P., Arefin, M. R., Misra, D., Charlin, L., & Rish, I. (2023). Challenging Common Assumptions about Catastrophic Forgetting and Knowledge Accumulation.

---

> > ### Comment · Reviewer_XNx6 · 2023-11-22
> > **Response to Authors**
> >
> > I thank the authors for carefully addressing my concerns. **I have read the author's comments** and was convinced by their answers. The new ablation studies shed light on the role of individual components and it seems to me that each extension is contributing to improving the performance. I still think that there should be a better way of addressing the scale difference of the weights at different layers but at the same time, the weight decay, in a sense tries to mitigate that problem. Thereby, I raised my initial score.

---

> > > ### Author Response · Authors · 2023-11-23
> > >
> > > We agree with the reviewer that scale is important and weight decay is likely helping here. We thank the reviewer for the insight and engaging in the discussion.

---

### Author Response · Authors · 2023-11-14

We would like to extend our sincere gratitude to all reviewers for the time and effort invested in reviewing our paper. We carefully considered each concern and suggestion and provided detailed responses.

We observe that reviewers have varied concerns, with no major common issue. To further support the contribution of utility gating, we have added additional ablation that was also requested by two reviewers (see ablation [[link]](https://drive.google.com/file/d/15aeYYq3QjeEgAbeFVeqzSNT5Ms2QvSYr/view?usp=sharing)).

**[Update]** We added a paper revision that includes 1) the new requested ablation study, 2) improved visualization (e.g., larger font size and better color contrast), 3) adding missing references, and 4) correcting the order of mention for Figures 6 and 7. The changes made in the text are blue-colored.

We eagerly await your responses and look forward to our discussions.

Sincerely,
\
The Authors

---

### Meta-Review · Area_Chair_hCas · 2023-12-18

**Metareview:**

This paper Utility-based Perturbed Gradient Descent (UPGD) which cleverly addresses the two important challenges in continual learning: forgetting and plasticity. The results in the standard streaming settings that are commonly used for plasticity research show that the proposed method reduces the loss of plasticity significantly.

One of the main criticisms from the reviewers was the lack of proper ablations. The authors provided ablations during the rebuttal which is very convincing. The reviewer who rejects the paper is 3zCE. However, their arguments are not valid. They complained about the lack of novelty and also about the experimental setup. I think this is mainly because the reviewer has not worked on plasticity research before. The setup is the standard one used in the literature and the solution is interesting.

I recommend an acceptance.

**Justification For Why Not Higher Score:**

This is a solid work but not at the spotlight level.

**Justification For Why Not Lower Score:**

No reason to reject.

---

### Decision · Program_Chairs · 2024-01-16

Accept (poster)